# Country-level instability is related to a stronger perceived climate of polarization across 44 countries
Amy S. G. Lee [1] ✉, Kelly Kirkland [1,2] & Brock Bastian [1]

Deep-seated partisan animosity and aversion present challenges across the global political landscape, yet little is known about how individuals perceive the climate of polarization in their societies or the broader societal conditions that shape these perceptions. Drawing on a dataset spanning 44 countries ($N = 8917$), we examined how macro-level indicators of dysfunction and disruption relate to the *perceived climate of polarization* – that is, the degree to which people believe major voter groups in their society dislike, distrust, and distance themselves from one another. We also tested whether perceptions of anomie in society – specifically, a perceived breakdown in the leadership and social fabric – may partially explain these relationships. Our analyses revealed stronger perceptions of partisan antipathy and distance in countries facing conditions linked to broader systemic instability, including high economic inequality, violence, weak governance, and a fragmented digital media landscape. Perceptions of anomie in society may partially explain these links. By contrast, no significant associations were found for environmental and public health indicators or the strength of democracy. These results suggest perceptions of a polarized societal climate may be linked to dysfunctional and disruptive economic, political, governmental, and social conditions that spark a felt sense of instability and disorder.

In 2024 and 2025, voters in more than 70 countries around the world participated in national elections that will reshape global politics for years to come. In many of these countries, there are concerns about deepening political divisions as societies seemingly devolve into adversarial political factions[1–3]. Escalating concerns about political polarization have motivated a growing body of literature investigating its origins and consequences. This research primarily focuses on two forms of polarization: *ideological polarization*, or the distance between partisans' policy or issue-based positions, and *affective polarization*, or the deep-seated animosity, distrust, and avoidance between political groups[4]. Findings underscore the harmful effects of affective polarization, in particular, which may undermine social cohesion, erode trust in democratic institutions, and hinder effective policy development and governance[5–9]. Recent work has also identified the specific conditions under which partisan animosity shapes political behaviors[10]. Yet relatively less attention has been given to how people perceive antagonistic dynamics between political groups, or the broader conditions that correspond with those perceptions.

The current study examines the *perceived climate of polarization*, which captures the degree to which people believe major voter groups in their society dislike, distrust, and distance themselves from one another. Rather than assessing personal feelings toward political outgroups, this perspective aims to understand how people perceive the broader relational dynamics between partisan groups in their society. This approach builds on research exploring perceived polarization, which has primarily focused on perceptions of ideological divisions. Past research suggests individuals often overestimate the ideological distance and demographic differences between voter groups in society[11,12] and these misperceptions can motivate more extreme policy positions[13] and amplify dislike and distrust for political opponents[14,15]. Some research has also examined beliefs about political outgroup animosity and distrust, capturing perceptions of affective divisions from a partisan perspective[4,16]. Together, this literature suggests that perceived polarization – the *psychological experience* of polarization – may shape how people think and behave within polarized contexts. By shifting the focus from partisan perspectives to broader perceptions of antipathy and avoidance between political groups, our approach opens further avenues for understanding how people make sense of political polarization in society.

Addressing perceptions of the societal climate rather than personal animus also contributes to existing literature on affective polarization in two

¹Melbourne School of Psychological Sciences, University of Melbourne, Melbourne, VIC, Australia. ²School of Psychology, University of Queensland, Brisbane, QLD, Australia. ✉e-mail: amylee@student.unimelb.edu.au

important ways. First, this approach allows us to explore how affective polarization is experienced by *everyone* in society, including political independents. Even individuals who do not personally express partisan bias may shape, experience, and be influenced by antagonistic dynamics between voter groups. Yet political independents are often omitted or reassigned a partisan identity in research relying on feeling thermometers that capture negative evaluations of political outgroups[17–20]. Second, this approach facilitates more inclusive multinational research by overcoming the challenges associated with aggregating partisan like-dislike scores in multiparty systems, where the number and structure of political groups can vary widely[19,21,22]. By accommodating both non-partisan perspectives and diverse political contexts, the perceived climate of polarization offers opportunities to further explore the social dimensions of polarization around the world.

How might broader societal conditions relate to perceptions of a polarized societal climate? One relevant lens comes from research on anomie, which captures the collective perception that society has become disintegrated and dysregulated due to weakened social cohesion and ineffective institutions[23–25]. In a 28-country multinational study, Teymoori and colleagues[24] found that societal instability was associated with heightened perceptions of breakdown in the social fabric and leadership. We propose that such perceptions can influence how people interpret and organize their social world. When people see society as unstable, they may adopt simpler, more structured interpretations of their social environment to regain a sense of predictability and control[25–27]. Establishing clear-cut political categories, in particular, can offer people a straightforward heuristic for anticipating others' values, beliefs, and behaviors. However, this process may also amplify perceived group differences[28,29], reinforcing a sense that political camps inhabit separate moral and social worlds. Indeed, prior research shows that perceptions of anomie are associated with lower perceived community cohesion[25]. In this way, perceptions of anomie may serve as a mechanism linking societal instability to heightened perceptions of partisan antipathy and distance. Testing these relationships could contribute a more nuanced understanding of how macro-level features of society relate to the perceived climate of polarization.

## The current study

Using a multinational dataset spanning 44 countries, the current study aimed to explore the societal conditions associated with a stronger perceived climate of polarization. A secondary goal was to consider whether perceptions of anomie could represent a psychological pathway linking societal conditions to people's impressions of the broader sociopolitical climate. To this end, we conducted exploratory analyses testing the direct relationships between dysfunctional and disruptive country-level conditions and the perceived climate of polarization. We then examined whether the same societal conditions are related to perceptions of anomie – specifically, a perceived breakdown in the leadership and social fabric. These analyses clarify whether the macro-level indicators correspond with subjective impressions of societal instability. We also evaluated whether perceptions of leadership and social fabric breakdown directly relate to the perceived climate of polarization. Finally, we conducted exploratory mediation analyses to assess whether perceived anomie may help explain links between adverse societal conditions and the perceived climate of polarization. As these analyses were conducted using observational data with unmanipulated mediators, they do not rule out alternative causal pathways and should be interpreted as a preliminary exploration of potential mechanisms rather than evidence of causality[30].

## Methods

### Participants and Procedure

The current study received ethics approval from the University of Melbourne ethics committee (Project #28441), and all methods were carried out in accordance with the relevant ethical regulations. The data for this study came from a larger survey involving measures that are not discussed here. On March 27, 2024, we pre-registered the data collection method and measures for the larger survey on the Open Science Framework (OSF;

https://osf.io/ghjdu/overview). However, we did not pre-register the aims and hypotheses of the current study due to its exploratory nature. Data, analysis code, and research materials are available on OSF.

Data were collected by an external survey company, Dynata, between May and June 2024. Based on the sample sizes obtained in past multinational datasets[31,32] and the cost of recruiting participants through Dynata, we aimed to recruit a minimum of 200 participants from 44 countries ($N = 8917$): Argentina, Australia, Austria, Belgium, Brazil, Bulgaria, Canada, Chile, China, Colombia, Czech Republic, Denmark, Finland, France, Germany, Greece, Hong Kong, Hungary, India, Indonesia, Ireland, Italy, Japan, Mexico, Netherlands, New Zealand, Norway, Philippines, Poland, Portugal, Republic of Korea, Romania, Singapore, Slovakia, South Africa, Spain, Sweden, Taiwan, Thailand, Türkiye, United Arab Emirates, United Kingdom, United States, and Vietnam. The 44 countries were selected based on availability through Dynata. As such, the sample represents a convenience sample. Dynata recruited participants through pre-established panels in each country, aiming for an equal gender split and a broad age distribution among adults aged 18 and older. However, strict quotas and population weights were not applied. Consequently, the samples should not be considered nationally representative. More information about the samples of each country can be found in Supplementary Tables 1 and 2.

All participants provided informed consent and were compensated monetarily for their participation. Participant ages ranged from 18-95 years ($M = 45.46$, $SD = 15.84$), and 51.8% identified as women, 48.0% as men, and 0.2% preferred another term or did not answer. Further details about gender reporting can be found in Supplementary Table 3. All participants passed two attention checks, and no participants were considered outliers for survey completion time according to the median time taken and interquartile range.

### Ethics and Inclusion Statement

This study involved data collected by an external survey company rather than researchers local to each of the countries sampled. We recognize the importance of local collaborators in research conducted in inhabited locations and have drawn on country-specific samples collected by local research teams in previous work[27,31,33]. However, time constraints and the scope of this project precluded direct collaboration with researchers in each of the 44 countries sampled. We instead worked with Dynata, which has established and ongoing relationships in those countries.

### Measures

An external company translated the survey items from English into the dominant language of each non-English-speaking country. We used OpenAI's Generative Pre-Trained Transformer (GPT; specifically, gpt-4-0314) to create back-translations, then manually verified these for errors. See Supplementary Table 3 for the exact wording of all survey items.

### Individual-level measures

Perceived climate of polarization. Participants were asked to think about the dominant groups of voters in their society and to assess the extent to which those groups: (1) dislike one another, (2) distrust one another, and (3) distance themselves from one another. These items reflect three social trends – dislike, distrust, and social distance or avoidance – that are commonly discussed as indicators of affective polarization[5,6,17,18]. By focusing on these salient dimensions of affective polarization, we aimed to assess perceptions of polarization in the social sphere, beyond ideological differences between major voter groups. Importantly, the phrasing of items did not limit participants to envisioning only two major voter groups – participants could answer the questions even if they perceived three, four, or more groups in their country. Participants were free to interpret the questions in a way that reflects the most salient cleavages in their national context. As a result, we were able to measure the perceived climate of polarization across diverse political landscapes, including multi-party systems. The three items ($\omega_{within} = 0.83$, $\omega_{between} = 0.99$) were rated on a scale from 1 (*Not at all*) to 5

(*Extremely*). We present details about construct validity and measurement invariance for this scale in Supplementary Note 1.

Anomie. Participants were asked to think about their society and to rate their level of agreement with 12 items from Teymoori and colleagues' Perceptions of Anomie scale[24]. The scale consists of two subscales pertaining to a perceived breakdown of the leadership and social fabric.

*Perceived Breakdown of Leadership.* Participants were asked to rate their level of agreement with six items (e.g., "Politicians don't care about the problems of the average person", "The government works towards the welfare of people"). The six items ($\omega_{within}$ = 0.81, $\omega_{between}$ = 0.94) were rated on a scale from 1 (*Strongly disagree*) to 7 (*Strongly agree*).

*Perceived Breakdown of Social Fabric.* Participants were asked to rate their level of agreement with six items (e.g., "People think that there are no clear moral standards to follow", "People are cooperative"). The six items ($\omega_{within}$ = 0.71, $\omega_{between}$ = 0.92) were rated on a scale from 1 (*Strongly disagree*) to 7 (*Strongly agree*).

Both subscales were included to capture people's broader interpretations of societal conditions. Although related measures such as external efficacy and trust in government may be more common in the literature, they emphasize narrower judgments about institutional responsiveness and legitimacy. In contrast, the Perceptions of Anomie scale encompasses evaluations of institutional breakdown alongside perceptions of erosion in the social fabric. The social fabric subscale reflects beliefs that moral standards are weak, that people are uncooperative and untrustworthy, and that interactions are not governed by shared norms. This broader perspective allows us to assess whether dysfunctional and disruptive societal conditions correspond with a diffuse sense of societal breakdown, beyond just discontent toward institutions. The Perceptions of Anomie scale therefore provides a more informative lens for examining whether and how societal conditions may relate to the perceived climate of polarization.

Demographic measures. In addition to questions about their age and gender identity, participants were asked to rate their religiosity ("If you follow a religion, how important is that religion in your daily life?") on a scale from 1 (*Not at all important*) to 7 (*Extremely important*). Participants were also asked to rate their socioeconomic status ("Relative to others in your country, how would you classify your own wealth?") on a scale from 1 (*Very poor*) to 7 (*Very wealthy*). Finally, participants responded to two items relating to political orientation on economic ("Please indicate your political beliefs from left/liberal to right/conservative on issues of the economy [e.g., social welfare, government spending, tax cuts]") and social issues ("Please indicate your political beliefs from left/liberal to right/conservative on social issues [e.g., immigration, homosexual marriage, abortion]"). Both items were rated on a scale from 1 (*Left/liberal*) to 7 (*Right/conservative*).

Country-level Measures. We sourced a variety of country-level measures from external databases to identify the macro-level societal conditions associated with a stronger perceived climate of polarization (see Table 1). We specifically selected indicators of societal dysfunction or disruption, which spanned (1) the economy and inequality, (2) violence and conflict, (3) environmental and public health, (4) governance, and (5) the digital media landscape. We included both "objective" indicators (e.g., gross domestic product [GDP] based on purchasing power parity [PPP] per capita) and expert assessments (e.g., political instability) to explore a greater variety of societal conditions in the analyses.

### Reporting summary
Further information on research design is available in the Nature Portfolio Reporting Summary linked to this article.

### Method of analysis
Data from the current study were collected from 44 samples. To account for the nesting of individuals (Level 1) within countries (Level 2), we conducted a series of linear mixed models (LMMs) – also referred to as hierarchical or

multilevel models – using Rstudio with the *lme4* package[34]. In all LMMs, we controlled for demographic measures, which were programmed as fixed effects, and country was modeled as a random intercept. Random slopes were not included, as many models failed to converge.

We began by examining relationships between demographic measures and the three perception variables: the perceived climate of polarization, perceived breakdown of leadership, and perceived breakdown of social fabric. We then centered and scaled all country-level indicators and assessed their effects on the perceived climate of polarization in separate LMMs that controlled for the demographic measures. These models aimed to identify the societal conditions associated with the perceived climate of polarization. Next, we explored the relationships between country-level indicators and perceptions of anomie, re-running the previous models twice with (1) perceived breakdown of leadership and (2) perceived breakdown of social fabric as the outcome variables. We then tested whether perceptions of leadership and social fabric breakdown directly related to the perceived climate of polarization in a separate LMM that controlled for demographics.

Finally, we conducted exploratory tests of indirect effects using the *lavaan* package[35] to examine whether the two dimensions of anomie mediated the relationship between each country-level indicator and the perceived climate of polarization. Each model included the anomie subscales in a parallel structure and controlled for the effects of the demographic measures. For comprehensiveness, mediation models were estimated for all country-level variables. However, we focus below on the country-level indicators that had significant direct associations with the perceived climate of polarization ($p < .05$). Full results are reported in Supplementary Table 6.

We also conducted a post hoc sensitivity power analysis to evaluate the minimum detectable effect size given our multilevel design. Based on 44 countries with approximately 200 participants per country (ICC = 0.12), a closed-form $z$-based approximation indicated that the models had 80% power ($\alpha$ = 0.05, two-tailed) to detect standardized country-level effects of $|\beta|$ = 0.15 or larger.

## Results
The pooled sample included 8917 participants from 44 countries. Summary statistics for demographic measures are presented in Table 2. Across all countries, participants reported moderate economic and social political orientations, as well as moderate socioeconomic status and religiosity. We present a breakdown of sample characteristics by country in Supplementary Table 1.

The intraclass correlation coefficient suggested that 12.5% of the variance in the perceived climate of polarization could be explained at the country level. Across all countries, the mean score for the perceived climate of polarization was slightly above the midpoint of the scale (M = 3.20, SD = 0.88). Meanwhile, 15.0% of the variance in the perceived breakdown of leadership and 10.6% of the variance in the perceived breakdown of social fabric could be explained at the country level. Across all countries, perceived leadership breakdown and perceived social fabric breakdown had moderate mean scores of 4.39 (SD = 1.18) and 4.43 (SD = 0.93), respectively.

As shown in Table 3, a stronger perceived climate of polarization was significantly associated with younger age and lower socioeconomic status. It was not significantly associated with gender, political orientation, or religiosity. Stronger perceptions of leadership breakdown tended to be reported by women and participants who were younger, less economically conservative, more socially conservative, less wealthy, and less religious. Meanwhile, stronger perceptions of social fabric breakdown tended to be reported by younger participants and those who were more socially conservative, less wealthy, and more religious.

Before interpreting the output of the LMMs featuring country-level predictors, we checked four key assumptions: linearity of relationships, independence of observations, normality of residuals, and homoscedasticity. Following recommendations by Fife[36], we took a graphical approach and drew diagnostic plots for each LMM using the *sjPlot* R package[37]. Visual inspection of these plots did not reveal any severe

## Table 1 | Descriptions and Sources of Country-Level Measures

| Measure | Description |
|---|---|
| ***Economy and inequality*** | |
| GDP PPP per capita[50] | Economic output measured in constant 2017 international dollars, accounting for the rate of inflation and relative cost of goods and services. |
| Economic inequality (Gini)[50] | Income inequality ranging from 0 (most equality) to 100 (most inequality). |
| Unemployment[51] | Percentage of the labor force that is jobless. |
| Youth not in employment, education, or training (NEET)[50] | Percentage of youth (aged 15–29 years) who are not in education, employment, or training. |
| ***Violence and conflict*** | |
| Conflict & instability[52] | Relative peacefulness ranging from 1 (most peaceful) to 4 (least peaceful), accounting for safety, security, and ongoing conflicts. |
| Homicide[53] | Intentional homicides per 100,000 population. |
| Political violence[54] | Use of political violence by non-state actors, with smaller values indicating lower frequency. |
| ***Environmental and public health*** | |
| Drought, floods & extreme temperatures[55] | Annual average percentage of the population who is impacted by natural disasters. |
| Pathogen prevalence[56] | Historical prevalence of infectious diseases, with values below zero indicating lower-than-mean prevalence. |
| Food insecurity[57] | Food affordability, availability, quality, safety, sustainability, and adaptation, with lower values indicating greater food insecurity. |
| Childhood mortality[50] | Mortality rate of children aged under five years per 1000 live births. |
| Life expectancy[58] | Average life expectancy at birth, measured in years. |
| ***Governance*** | |
| Strength of democracy[59] | Relative strength of democracy ranging from 0 (least democratic) to 10 (most democratic), accounting for electoral processes, government functioning, political participation, political culture, and civil liberties. |
| Political stability[60] | Perceived likelihood of political instability and/or politically motivated violence, ranging from −2.5 (weak governance) to 2.5 (strong governance). |
| Government effectiveness[60] | Perceived credibility of government policy commitments and quality of public services, civil service, and policy development and implementation, ranging from −2.5 (weak governance) to 2.5 (strong governance). |
| Rule of law[60] | Perceived confidence in and abidance by societal rules, ranging from −2.5 (weak governance) to 2.5 (strong governance). |
| Corruption control[60] | Perceived use of public power for private gain, ranging from −2.5 (weak governance) to 2.5 (strong governance). |
| ***Digital media landscape*** | |
| Daily time spent on internet[61] | Average daily minutes spent on the internet by users aged 16–64 years. |
| Daily time spent on social media[61] | Average daily minutes spent on social media by users aged 16−64 years. |
| Online media consistency[54] | Consistency in news presentation across major domestic online media outlets, with smaller values indicating reduced consistency. |

## Table 2 | Demographic details across the full sample

| Variable | *M* | *SD* | Range |
|---|---|---|---|
| Age | 45.46 | 15.84 | 18–95 |
| Economic political orientation | 3.98 | 1.55 | 1–7 |
| Social political orientation | 3.94 | 1.65 | 1–7 |
| Socioeconomic status | 3.84 | 1.13 | 1–7 |
| Religiosity | 3.55 | 2.20 | 1–7 |
| **Gender** | ***N*** | **Percentage** | |
| Female | 4622 | 51.8 | |
| Male | 4277 | 48.0 | |
| Other/Prefer not to say | 18 | 0.2 | |

violations of the four assumptions. Minor deviations were not addressed as LMMs are relatively robust to violations of distributional assumptions[38].

As detailed in Table 4, the LMMs featuring country-level indicators found a stronger perceived climate of polarization in countries with less favorable economic conditions (lower GDP PPP per capita; greater income inequality, unemployment, and youth NEET), as well as heightened violence and conflict (greater conflict and instability, political violence, and

homicide). Governance-related indicators were the strongest correlates: a one-standard-deviation increase in government effectiveness corresponded with a 0.20-point lower score on the perceived climate of polarization, with similar associations observed for corruption control (−0.18) and rule of law (−0.17). For youth NEET and conflict and instability, a one-standard-deviation increase was associated with a ~0.15-point higher score on the perceived climate of polarization. Relatively weaker, but statistically significant, associations were also observed for GDP PPP per capita, income inequality, homicide, and political violence.

Indicators of the digital media landscape were also significantly associated with perceived polarization, though effect sizes were smaller in magnitude ($|\beta| < 0.15$). Lower consistency in online media and higher internet and social media use were each associated with a stronger perceived climate of polarization. By contrast, environmental and public health indicators (droughts, floods, and extreme temperatures; pathogen prevalence; food insecurity; childhood mortality; and life expectancy) and democratic strength showed non-significant and near-zero relationships.

As shown in Table 4, many of the country-level indicators associated with the perceived climate of polarization were also significantly related to perceptions of anomie, with more consistent associations observed for the social fabric subscale. Higher income inequality, unemployment, and youth NEET were positively related to both anomie subscales, with unemployment showing the strongest association with perceived breakdown of

**Table 3 | Linear mixed models examining the effects of demographic variables on the perceived climate of polarization, perceived breakdown of leadership, and perceived breakdown of social fabric**

| Predictors | Perceived climate of polarization | | | | Perceived breakdown in leadership | | | | Perceived breakdown in social fabric | | | |
|---|---|---|---|---|---|---|---|---|---|---|---|---|
| | β | df | 95% CI | p | β | df | 95% CI | p | β | df | 95% CI | p |
| (Intercept) | 3.21 | 46.42 | [3.11, 3.30] | <.001*** | 4.33 | 46.02 | [4.20, 4.46] | <.001*** | 4.41 | 47.00 | [4.32, 4.50] | <.001*** |
| Age | −0.02 | 8516.76 | [−0.04, 0.00] | .016* | −0.08 | 8427.71 | [−0.11, −0.06] | <.001*** | −0.12 | 8416.06 | [−0.14, −0.10] | <.001*** |
| Gender [female] | −0.02 | 8507.91 | [−0.06, 0.02] | .285 | 0.10 | 8419.56 | [0.05, 0.14] | <.001*** | 0.04 | 8406.50 | [0.00, 0.07] | .061 |
| Economic political orientation | −0.01 | 8515.91 | [−0.03, 0.02] | .623 | −0.04 | 8426.65 | [−0.07, 0.00] | .044* | 0.01 | 8415.61 | [−0.01, 0.04] | .344 |
| Social political orientation | 0.00 | 8514.57 | [−0.03, 0.02] | .814 | 0.10 | 8425.67 | [0.07, 0.14] | <.001*** | 0.04 | 8414.21 | [0.01, 0.07] | .004** |
| Socioeconomic status | −0.04 | 8540.98 | [−0.05, −0.02] | <.001*** | −0.23 | 8450.25 | [−0.25, −0.21] | <.001*** | −0.10 | 8442.12 | [−0.12, −0.08] | <.001*** |
| Religiosity | 0.00 | 8502.10 | [−0.02, 0.02] | .819 | −0.09 | 8432.69 | [−0.12, −0.07] | <.001*** | 0.02 | 8363.43 | [0.00, 0.05] | .028* |
| *Random Effects* | | | | | | | | | | | | |
| Residual | 0.68 | | | | 1.12 | | | | 0.75 | | | |
| Country (intercept) | 0.10 | | | | 0.18 | | | | 0.09 | | | |
| ICC | 0.124 | | | | 0.14 | | | | 0.11 | | | |
| Observations | 8555 | | | | 8467 | | | | 8453 | | | |
| Marginal $R^2$ | 0.003 | | | | 0.056 | | | | 0.029 | | | |
| Conditional $R^2$ | 0.126 | | | | 0.186 | | | | 0.133 | | | |

Separate models were programmed with (1) the perceived climate of polarization, (2) perceived breakdown of leadership, and (3) perceived breakdown of social fabric as outcome variables. Gender was coded as male (1) and female (2). Marginal $R^2$ only pertains to fixed effects and Conditional $R^2$ pertains to the entire model. Predictors have been standardized ($M = 0$, $SD = 1$), so beta values represent expected change in the perception variables (in raw units) for one $SD$ increase in the predictor. *CI* confidence interval. *$p < .05$, **$p < .01$, ***$p < .001$.

leadership (β = −0.20, p = .001) and social fabric (β = 0.17, p < .001). Higher homicide rates were also associated with both leadership (β = 0.19, p = .003) and social fabric breakdown (β = 0.14, p = .002), whereas political violence exhibited relatively weaker (|β| <0.15), though significant, relationships.

Of the governance indicators, only government effectiveness and corruption control were consistently related to both anomie subscales; weaker political stability and rule of law were only significantly associated with stronger perceptions of social fabric breakdown. Similar results emerged for the digital media landscape: online media consistency was significantly associated with both subscales, whereas internet and social media use showed weaker but significant associations with only social fabric breakdown (|β| < 0.15). By contrast, environmental and public health indicators showed an inconsistent pattern of associations with perceived leadership and social fabric breakdown, and democratic strength was not significantly related to either anomie subscale.

The broader pattern of results across models suggested that similar macro-level conditions are generally linked to both perceptions of anomie and beliefs about pervasive political divisions. We next examined whether perceptions of anomie were directly associated with the perceived climate of polarization in a separate model. Results indicated significant positive associations for both anomie subscales when entered simultaneously into the model. Specifically, perceptions of leadership breakdown were positively associated with the perceived climate of polarization (β = 0.07, *SE* = 0.01, p < .001, 95% CI [0.05, 0.09]), while perceptions of social fabric breakdown showed an even stronger positive relationship (β = 0.26, *SE* = 0.01, p < .001, 95% CI [0.24, 0.28]).

Building on these direct associations, we next explored whether perceptions of anomie could help explain the link between country-level conditions and the perceived climate of polarization. As detailed in Table 5, parallel mediation models showed that the significant associations between country-level indicators and the perceived climate of polarization were partly explained by perceptions of breakdown in the social fabric and, to a lesser extent, the leadership. For several indicators – including GDP PPP per capita, conflict and instability, political stability, rule of law, daily internet use, and daily social media use – only perceptions of social fabric breakdown partially explained shared variance with perceived polarization. For example, a one-standard-deviation decrease in GDP PPP per capita was associated with a 0.13-point higher perceived polarization score, of which 23.1% was statistically attributable to perceptions of the social fabric. Meanwhile, the indirect pathway through perceived leadership breakdown was near-zero and non-significant. Across these country-level indicators, a perceived breakdown in the social fabric explained 27.5–30.8% of the overall association with the perceived climate of polarization.

For other indicators, perceptions of both social fabric and leadership breakdown partially explained associations with the perceived climate of polarization. Income inequality, unemployment, youth NEET, homicide, political violence, government effectiveness, corruption control, and online media consistency had significant indirect pathways through both mediators, though the contribution of social fabric perceptions was consistently larger. For example, 8.3% of the association between income inequality and the perceived climate of polarization could be explained via perceptions of leadership breakdown, compared to 33.3% via perceptions of social fabric breakdown. Similarly, a perceived breakdown in the social fabric accounted for 30.8% of the association between homicide and perceived polarization, while perceived leadership breakdown only accounted for 7.7%.

## Discussion

Deep-seated partisan animosity and aversion pose a growing threat to the healthy functioning of societies around the world[1–3]. However, limited research has explored how people perceive the depth and intensity of political divisions in their societies, as well as the broader macro-level conditions associated with these views. Through exploratory analyses on a multinational dataset spanning 44 countries, we aimed to gain a clearer picture of the societal conditions associated with stronger perceptions of a polarized climate. We also examined whether perceptions of anomie are related to these societal conditions and to perceived polarization. Analyses

**Table 4 | Linear mixed models examining the effect of country-level variables on the perceived climate of polarization, perceived breakdown of leadership, and perceived breakdown of social fabric**

| Predictors | Perceived climate of Polarization | | | | | Perceived breakdown of leadership | | | | | Perceived breakdown of social fabric | | | | |
|---|---|---|---|---|---|---|---|---|---|---|---|---|---|---|---|
| | N | β | df | 95% CI | p | N | β | df | 95% CI | p | N | β | df | 95% CI | p |
| **Economy and inequality** | | | | | | | | | | | | | | | |
| GDP PPP per capita | 8360 | −0.13^ | 41.14 | [−0.22, −0.04] | .005 ** | 8277 | −0.11^ | 41.05 | [−0.24, 0.02] | .100 | 8260 | −0.11^ | 41.14 | [−0.20, −0.02] | .018 * |
| Income inequality | 7783 | 0.12^ | 38.28 | [0.03, 0.21] | .012 * | 7707 | 0.16 | 38.14 | [0.03, 0.28] | .018 * | 7692 | 0.13^ | 38.34 | [0.04, 0.22] | .005 ** |
| Unemployment | 8555 | 0.14^ | 42.01 | [0.06, 0.23] | .002 ** | 8467 | 0.20 | 41.88 | [0.09, 0.32] | .001 ** | 8453 | 0.17 | 41.91 | [0.09, 0.25] | <.001 *** |
| Youth NEET | 8170 | 0.15 | 40.42 | [0.06, 0.24] | .002 ** | 8089 | 0.14^ | 40.25 | [0.02, 0.27] | .032 * | 8074 | 0.17 | 40.62 | [0.09, 0.25] | <.001 *** |
| **Violence and conflict** | | | | | | | | | | | | | | | |
| Conflict & instability | 8365 | 0.15 | 41.39 | [0.07, 0.24] | <.001 *** | 8280 | 0.10^ | 41.25 | [−0.03, 0.22] | .147 | 8262 | 0.14^ | 41.54 | [0.06, 0.23] | .002 ** |
| Homicide | 8360 | 0.13^ | 41.16 | [0.04, 0.21] | .009 ** | 8277 | 0.19 | 41.02 | [0.07, 0.31] | .003 ** | 8260 | 0.13^ | 41.17 | [0.04, 0.21] | .006 ** |
| Political violence | 8555 | 0.10^ | 42.16 | [0.01, 0.19] | .030 * | 8467 | 0.13^ | 42.07 | [0.01, 0.25] | .042 * | 8453 | 0.10^ | 42.17 | [0.02, 0.19] | .025 * |
| **Environmental and public health** | | | | | | | | | | | | | | | |
| Drought, floods & extreme temperatures | 7970 | −0.06^ | 39.05 | [−0.15, 0.04] | .262 | 7896 | −0.21 | 39.04 | [−0.32, −0.10] | <.001 *** | 7880 | −0.07^ | 38.95 | [−0.16, 0.02] | .141 |
| Pathogen prevalence | 8364 | 0.02^ | 41.28 | [−0.07, 0.12] | .614 | 8272 | −0.07^ | 41.18 | [−0.20, 0.06] | .322 | 8258 | 0.04^ | 41.28 | [−0.06, 0.13] | .451 |
| Food insecurity | 8170 | −0.09^ | 40.56 | [−0.19, 0.00] | .058 | 8090 | −0.06^ | 40.42 | [−0.19, 0.07] | .397 | 8069 | −0.12^ | 40.74 | [−0.21, −0.03] | .010 ** |
| Childhood mortality | 8170 | 0.02^ | 40.46 | [−0.08, 0.12] | .673 | 8090 | 0.01^ | 40.34 | [−0.12, 0.14] | .873 | 8069 | 0.07^ | 40.55 | [−0.02, 0.17] | .126 |
| Life expectancy | 8170 | −0.06^ | 40.45 | [−0.16, 0.03] | .215 | 8090 | −0.05^ | 40.33 | [−0.18, 0.09] | .493 | 8069 | −0.10^ | 40.56 | [−0.19, −0.01] | .039 * |
| **Governance** | | | | | | | | | | | | | | | |
| Strength of democracy | 8555 | −0.06^ | 42.23 | [−0.15, 0.04] | .244 | 8467 | 0.13^ | 42.16 | [0.00, 0.25] | .051 | 8453 | −0.05^ | 42.23 | [−0.14, 0.04] | .260 |
| Political stability | 8555 | −0.15 | 42.40 | [−0.23, −0.07] | .001 ** | 8467 | −0.08^ | 42.26 | [−0.21, 0.04] | .195 | 8453 | −0.14^ | 42.54 | [−0.22, −0.06] | .001 ** |
| Government effectiveness | 8555 | −0.20 | 42.34 | [−0.28, −0.13] | <.001 *** | 8467 | −0.17 | 42.19 | [−0.29, −0.06] | .006 ** | 8453 | −0.18 | 42.51 | [−0.26, −0.11] | <.001 *** |
| Rule of law | 8555 | −0.17 | 42.42 | [−0.25, −0.09] | <.001 *** | 8467 | −0.19 | 42.26 | [−0.23, 0.02] | .116 | 8453 | −0.15 | 42.56 | [−0.23, −0.07] | <.001 *** |
| Corruption control | 8555 | −0.18 | 42.40 | [−0.25, −0.10] | <.001 *** | 8467 | −0.14^ | 42.24 | [−0.26, −0.01] | .035 * | 8453 | −0.16 | 42.56 | [−0.24, −0.09] | <.001 *** |
| **Digital media landscape** | | | | | | | | | | | | | | | |
| Daily time spent on internet | 8169 | 0.12^ | 40.49 | [0.03, 0.21] | .010 ** | 8084 | 0.09^ | 40.31 | [−0.04, 0.22] | .170 | 8069 | 0.13^ | 40.60 | [0.04, 0.22] | .005 ** |
| Daily time spent on social media | 8169 | 0.12^ | 40.50 | [0.03, 0.21] | .016 * | 8084 | 0.07^ | 40.33 | [−0.06, 0.20] | .292 | 8069 | 0.13^ | 40.63 | [0.05, 0.22] | .005 ** |
| Online media consistency | 8555 | −0.14^ | 42.08 | [−0.22, −0.05] | .003 ** | 8467 | −0.19 | 42.01 | [−0.31, −0.08] | .002 ** | 8453 | −0.11^ | 42.01 | [−0.19, −0.02] | .017 * |

Each line indicates three separate LMMs with country-level variables predicting the (1) perceived climate of polarization, (2) perceived breakdown of leadership, and (3) perceived breakdown of social fabric. Predictors have been standardized ($M = 0$, $SD = 1$), so beta values represent expected change in the outcome variable (in raw units) for one $SD$ increase in the predictor. All models control for demographic controls (see Supplementary Table 5 for model estimates without demographic controls). ^ indicates beta values below 0.15, which were estimated with lower precision given the available power. $*p < .05$, $**p < .01$, $***p < .001$. CI confidence interval.

**Table 5 | Bootstrapped mediation effect of country-level measures on the perceived climate of polarization via perceptions of anomie**

| Predictor | N | Direct Effect | | | | Indirect Effect (Leadership Breakdown) | | | | Indirect Effect (Social Fabric Breakdown) | | | | Total Effect | | | |
|---|---|---|---|---|---|---|---|---|---|---|---|---|---|---|---|---|---|
| | | Est. | SE | 95% CI | p | Est. | SE | 95% CI | p | Est. | SE | 95% CI | p | Est. | SE | 95% CI | p |
| **Economy and inequality** | | | | | | | | | | | | | | | | | |
| GDP PPP per capita | 8422 | −0.09 | 0.03 | [−0.15, −0.04] | .001 ** | −0.01 | 0.01 | [−0.02, 0.00] | .150 | −0.03 | 0.01 | [−0.05, −0.01] | .003 ** | −0.13 | 0.04 | [−0.21, −0.06] | <.001 *** |
| Income inequality | 7842 | 0.07 | 0.03 | [0.01, 0.14] | .024 * | 0.01 | 0.00 | [0.00, 0.02] | .001 ** | 0.04 | 0.01 | [0.02, 0.06] | <.001 *** | 0.12 | 0.04 | [0.05, 0.20] | .001 ** |
| Unemployment | 8618 | 0.08 | 0.05 | [−0.02, 0.17] | .111 | 0.02 | 0.00 | [0.01, 0.02] | <.001 *** | 0.05 | 0.01 | [0.03, 0.08] | <.001 *** | 0.14 | 0.06 | [0.02, 0.26] | .019 * |
| Youth NEET | 8231 | 0.10 | 0.04 | [0.03, 0.17] | .008 ** | 0.01 | 0.01 | [0.00, 0.02] | .012 * | 0.05 | 0.01 | [0.04, 0.07] | <.001 *** | 0.16 | 0.04 | [0.08, 0.24] | <.001 *** |
| **Violence and conflict** | | | | | | | | | | | | | | | | | |
| Conflict & instability | 8425 | 0.11 | 0.02 | [0.07, 0.15] | <.001 *** | 0.01 | 0.01 | [0.00, 0.02] | .065 | 0.05 | 0.01 | [0.02, 0.07] | <.001 *** | 0.17 | 0.03 | [0.11, 0.22] | <.001 *** |
| Homicide | 8422 | 0.07 | 0.03 | [0.02, 0.13] | .009 ** | 0.01 | 0.00 | [0.01, 0.02] | <.001 *** | 0.04 | 0.01 | [0.03, 0.05] | <.001 *** | 0.13 | 0.03 | [0.07, 0.18] | <.001 *** |
| Political violence | 8618 | 0.06 | 0.02 | [0.01, 0.11] | .010 * | 0.01 | 0.00 | [0.00, 0.02] | .021 * | 0.03 | 0.01 | [0.01, 0.05] | .004 ** | 0.10 | 0.04 | [0.04, 0.17] | .003 ** |
| **Governance** | | | | | | | | | | | | | | | | | |
| Political stability | 8618 | −0.11 | 0.03 | [−0.17, −0.05] | <.001 *** | −0.01 | 0.01 | [−0.02, 0.01] | .229 | −0.04 | 0.01 | [−0.07, 0.02] | .002 ** | −0.16 | 0.04 | [−0.24, −0.08] | <.001 *** |
| Government effectiveness | 8618 | −0.15 | 0.02 | [−0.19, −0.10] | <.001 *** | −0.01 | 0.00 | [−0.02, −0.01] | .002 ** | −0.06 | 0.01 | [−0.07, −0.04] | <.001 *** | −0.21 | 0.03 | [−0.27, −0.16] | <.001 *** |
| Rule of law | 8618 | −0.13 | 0.03 | [−0.18, −0.07] | <.001 *** | −0.01 | 0.01 | [−0.02, 0.00] | .104 | −0.05 | 0.01 | [−0.07, −0.02] | .001 ** | −0.18 | 0.04 | [−0.26, −0.11] | <.001 *** |
| Corruption control | 8618 | −0.13 | 0.03 | [−0.18, −0.08] | <.001 *** | −0.01 | 0.01 | [−0.02, 0.00] | .023 * | −0.05 | 0.01 | [−0.07, −0.03] | <.001 *** | −0.19 | 0.04 | [−0.26, −0.12] | <.001 *** |
| **Digital media landscape** | | | | | | | | | | | | | | | | | |
| Daily time spent on internet | 8228 | 0.09 | 0.03 | [0.02, 0.15] | .007 ** | 0.01 | 0.01 | [0.00, 0.02] | .097 | 0.04 | 0.01 | [0.02, 0.06] | <.001 *** | 0.14 | 0.04 | [0.06, 0.21] | <.001 *** |
| Daily time spent on social media | 8228 | 0.08 | 0.04 | [0.01, 0.15] | .025 * | 0.01 | 0.01 | [0.00, 0.02] | .160 | 0.04 | 0.01 | [0.02, 0.07] | <.001 *** | 0.13 | 0.04 | [0.04, 0.21] | .003 ** |
| Online media consistency | 8618 | −0.09 | 0.03 | [−0.15, −0.03] | .002 ** | −0.01 | 0.01 | [−0.02, 0.00] | .005 ** | −0.03 | 0.01 | [−0.06, −0.01] | .013 * | −0.14 | 0.04 | [−0.21, −0.06] | <.001 *** |

Bootstrapped estimates after 5000 simulations. All models control for demographic variables (see Supplementary Table 7 for model estimates without demographic controls). Est. estimate. CI confidence interval. *p < .05, **p < .01, ***p < .001.

revealed a consistent pattern: perceptions of a polarized societal climate tended to be stronger in countries facing greater dysfunction and disruption across indicators of the economy and inequality, violence and conflict, governance, and the digital media landscape. Importantly, many of the same societal conditions were also associated with perceived anomie, which was in turn positively related to the perceived climate of polarization. By contrast, significant relationships were not consistently observed for environmental or public health indicators, nor for the strength of democracy.

Our findings suggest that perceptions of a polarized climate are linked to broader impressions of societal breakdown. Both perceived polarization and perceptions of erosion in the leadership and social fabric were elevated in countries characterized by unfavorable economic circumstances, heightened threats of violence, weaker governance, and more disruptive digital media environments. Positive associations between the anomie subscales and the perceived climate of polarization indicate that people who view their society as poorly governed or socially fragmented are also more likely to perceive antagonistic dynamics between political groups. Notably, perceptions of breakdown in the social fabric showed more consistent associations with adverse societal conditions and a stronger relationship with perceived polarization, suggesting that diffuse concerns about social cohesion may be particularly relevant to how people interpret political divisions. Taken together, these findings point to perceived anomie as one plausible psychological pathway linking adverse societal conditions to beliefs about partisan antipathy and distance. Results from the exploratory mediation analyses were consistent with this interpretation, though causal direction cannot be established.

We interpret these findings through a theoretical lens that links societal instability to a heightened psychological need to impose order and structure on the social environment. When societal instability takes the form of economic strain, violence, or ineffective governance, people may come to believe that society is unraveling. Such contexts can weaken ties and community engagement[39,40] and give rise to a sense of uncertainty and insecurity. In response, individuals may seek to restore a sense of control by imposing order and structure on their social environments[25–27,41]. Drawing sharper boundaries between political camps may help individuals navigate an unstable world by providing a powerful shorthand for interpreting others' values, beliefs, and behaviors. However, this approach may also reinforce perceived differences between groups[28,29], giving rise to the belief that they stand on opposite sides of a moral and social divide.

In addition to broader instability related to the economy, violence, and governance, disruption in the digital media landscape was also significantly associated with the perceived climate of polarization. People reported greater partisan antipathy and distance in countries with higher internet and social media use, as well as more fragmented online media environments. These same conditions were also linked to heightened perceptions of anomie, particularly a perceived breakdown in the social fabric. Prior research suggests digital media environments can amplify emotionally charged and divisive content[42–44], making political conflict appear more widespread and socially normative than it truly is[45,46]. In fragmented media ecosystems, people may also encounter conflicting accounts of reality, frequent derogation of political opponents, and a strong focus on political conflict[47]. Such content can deepen misperceptions about political outgroups[48] and foster impressions of chaos, weak leadership, and fractured norms. In turn, these perceptions may further entrench the belief that society is split into irreconcilable political factions.

Taken together, the macro-level indicators that were significantly associated with the perceived climate of polarization point to conditions of societal dysfunction and disruption that may fuel a *felt sense* of instability and disorder in everyday life. Many of these same conditions were also associated with heightened perceptions of anomie, reinforcing the idea that perceived polarization may be related to collective beliefs about societal breakdown. Notably, these indicators reflect economic, political, governmental, and social disturbances that are largely human-driven, rather than the result of seemingly uncontrollable external forces. By contrast, the perceived climate of polarization was not significantly related to

environmental or public health indicators, nor to democratic strength. The anomie subscales similarly showed inconsistent relationships with environmental and public health conditions and were not significantly associated with democratic strength. However, it is important to note that null results may reflect limited statistical power rather than the absence of a relationship, as our study was underpowered to reliably detect smaller effects. Future research with larger samples would be better placed to further explore these relationships.

The current work contributes to a growing literature that seeks to understand the emergence of affective polarization around the world[20,49]. By focusing on people's *perceptions* of interparty hostility and aversion and by systematically examining societal dysfunction, disruption, and perceived anomie, our work offers an additional lens for understanding how individuals may interpret broader societal conditions and how these perceptions may relate to beliefs about political divisions. Moreover, we have presented the first cross-national analysis of how people perceive the *climate* of partisan animosity in their society, using a large and diverse multinational dataset that includes the perspectives of partisans and non-partisans alike. Our approach offers opportunities to deepen our understanding of how people experience and make sense of affective polarization around the world.

### Limitations and future directions

Despite these strengths, we acknowledge several important limitations in the current research. First, because our design had 80% power to detect standardized country-level effects of $|\beta| = 0.15$ or larger, effects smaller than this threshold should be interpreted with caution, as the study was not optimally powered to reliably detect them. Additionally, our cross-sectional and exploratory design precludes strong conclusions about the directionality of relationships. While our mediation models explore a potential mechanism through which societal dysfunction and disruption may relate to the perceived climate of polarization, they provide no definitive basis for causal inference. As noted by Bullock and colleagues[30], mediation models with unmanipulated mediators are vulnerable to biased estimates and cannot account for all possible confounds. The results of our mediation analyses should therefore be interpreted with caution. Longitudinal or experimental studies are needed to further probe causal pathways.

Furthermore, while we aimed to explore a variety of potential country-level correlates of the perceived climate of polarization, our analyses were non-exhaustive and many of the country-level variables examined are likely interrelated. Future research should explore the role of other societal conditions, as well as how different factors may interact to shape perceptions of hostility and aversion between political groups. Finally, our country samples were not representative of population demographics, limiting the generalizability of findings. Moreover, while our sample included a broad and diverse set of countries, it did not include contexts facing the most acute forms of societal instability, such as active conflict zones or states experiencing total governmental collapse. Including such countries in future research could enhance our understanding of how extreme societal dysfunction and disruption relate to polarization. However, it is important to consider the practical and ethical constraints of conducting psychological research in such environments.

### Conclusion

Concerns about affective polarization are no longer confined to a handful of nations – they have become a concerning feature of the global political landscape. However, we still know little about how people perceive the intensity of partisan divisions in their societies and the broader macro-level conditions associated with these views. Drawing on data from 44 culturally diverse countries, our exploratory analyses found that indicators of societal instability – including unfavorable economic conditions, heightened violence, weaker governance, and more disruptive digital media environments – were linked to stronger perceptions of a polarized societal climate. Many of these same conditions were also associated with stronger perceptions of breakdown in the leadership and social fabric of society, which were, in turn,

**Article**

linked with heightened perceived polarization. Together, these findings indicate that perceptions of a polarized societal climate may be more likely to emerge in contexts characterized by a felt sense of instability and breakdown, particularly when societal challenges appear to result from systemic economic, political, governmental, and social dysfunction. Although further research is needed to clarify causal pathways, these results suggest beliefs about sociopolitical fragmentation may take root in the cracks of weakened systems. Further examining these dynamics may help identify the conditions under which societies become especially vulnerable to beliefs about pernicious political polarization.

## Data availability
Data are publicly available on OSF.

## Code availability
All analysis code is publicly available on OSF.

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

## Acknowledgements

This work was supported by research funds provided to Brock Bastian by the Melbourne School of Psychological Sciences at the University of Melbourne, as well as by an Australian Government Research Training Program Scholarship (https://doi.org/10.82133/C42F-K220) awarded to Amy S. G. Lee by the Commonwealth of Australia. The funders had no role in the study design, data collection and analysis, decision to publish, or manuscript preparation.

## Author contributions

A.S.G.L. contributed to conceptualization, methodology, data curation, formal analysis, investigation, writing (original draft preparation), and project administration. K.K. contributed to conceptualization, methodology, data curation, investigation, writing (review and editing), project administration, and supervision. B.B. contributed to conceptualization, methodology, investigation, writing (review and editing), project administration, supervision, and funding acquisition.

## Competing interests

The authors declare no competing interests.
