## [Transparent Peer Review file · Communications Psychology]

Country-level instability is related to a stronger perceived climate of polarization across 44 countries

Corresponding Author: Ms Amy Lee

Version 0:

Decision Letter:

Dear Ms Lee,

Thank you for your patience during the peer-review process. Your manuscript titled "Exploring the perceived climate of polarization across 44 countries in relation to broader societal conditions" has now been seen by 3 reviewers, whose comments are appended below. You will see that they find your work of some potential interest. However, they have raised quite substantial concerns that must be addressed. In light of these comments, we cannot accept the manuscript for publication, but would be interested in considering a revised version that fully addresses these serious concerns.

We hope you will find the Reviewers' comments useful as you decide how to proceed. Should additional work allow you to address these criticisms, we would be happy to look at a substantially revised manuscript. If you choose to take up this option, please highlight all changes in the manuscript text file, and provide a detailed point-by-point reply to the reviewers.

Editorially, we consider it critical that you address all of the referees' methodological concerns.

Moreover, we ask that you provide a sensitivity power analysis to demonstrate the (nested) samples' power to detect the smallest effect size of theoretical significance, with a scientific justification for this target effect size. We further ask that you avoid any directional or causal claims and clarify throughout that the work is exploratory (hypotheses and analyses were not preregistered).

I am attaching a checklist that details critical reporting requirements for the revised manuscript. Please attend to each item and ensure your manuscript is fully compliant. We are requesting that your manuscript aligns with these requirements as this facilitates the evaluation of your manuscript, reducing delays in re-review and potential future acceptance. If your revised manuscript is not aligned with these requests on major issues, such as those concerning statistics, it may be returned to you for further revisions without re-review. Additional information can be found in our style and formatting guide Communications Psychology formatting guide.

If the revision process takes significantly longer than five months, we will be happy to reconsider your paper at a later date, provided it still presents a significant contribution to the literature at that stage.

Please use the following link to submit your
- revised manuscript,
- point-by-point response to the referees' comments,

- cover letter (as a separate document),
- the Reporting Summary (see below), and
- the completed Editorial Request Table (attached):

Link Redacted

Thank you for the opportunity to review your work.

Best regards,
Marika Schiffer, on behalf of

Philipp Schmid

Philipp Schmid
External Editor
Communications Psychology

REVIEWER REPORTS:

Reviewer #1 (Remarks to the Author):

I enjoyed this paper. There have been a lot of “negative” trends in the world such as increasing economic inequality, decreasing efficacy, decreased well-being, increased social media usage / inattentiveness. It seems intuitive that these dynamics relate to perceptions of affective polarization. I also agree that they seem more relevant to perceptions than actual affective polarization and hence the focus here makes sense.

I only have a few comments (which is unusual – I typically have more to say but this strikes me as an important, very interesting, correlational demonstration).

First, the first paragraph connects affective polarization to the violation of democratic norms. This is an area of debate and at least the authors should footnote that, citing Broockman et al. 2022 (AJPS), Voelkel et al. 2023 (NHB), 2024 (Science), and Druckman et al. (book).

Second, the authors mention the exclusion of independents in many analyses. That’s a fair point although see Ahn and Mutz’s 2024 (POQ) paper. Regardless, they cite 43% of Americans as independents. I assume that this percentage is counting independent leaners as independents – and nearly all studies actually treat such respondents as partisans. Thus, I think it is likely an overstatement of how many people are excluded from these studies.

Third, the authors might tie their work to perceptions of polarization – even though much of that focuses on ideology: Levendusky and Malhotra 2016 (Poli Comm), Druckman et al. 2022 (JOP).

Fourth, the authors argue that understanding perceptions of polarization provides insight into how polarization develops. I’m not sure about the nature of that connection and it seems tangential to the point of the paper. Without more evidence and logic, I would just drop that point.

Fifth, the authors might find a recent study by Stapleton and Wolak 2024 (POQ) on efficacy and affective polarization. It is a similar conceptual argument I think.

Sixth, I appreciate the care put in to the measure and the including / checking back translations. I also found the results sensible/fascinating and carefully presented. I had some concerns about causal direction, but the authors thoughtfully address this in the conclusion. The appendix is also useful.

Finally, there are three analysis questions I have: 1) the mediation analysis is fine given norms in psychology from what I can gather but there is substantial debate in political science about this type of mediation analysis (see papers by John Bullock) – the author might acknowledge that the mediation cannot be definitive, 2) I would be interested in whether there are differences between WEIRD and non-WEIRD countries given that was an initial sampling motivation, and 3) I am not sure if this is too complicated but is there a way to look at any differences by based the government’s ideology complexion (left/right)?

Overall, these are small comments. I really enjoyed the paper.

Reviewer #2 (Remarks to the Author):

It was a pleasure to read "Exploring the perceived climate of polarization across 44 countries in relation to broader societal

conditions." In an exploratory fashion, the authors seek to answer an important question: why are countries likely to experience high (or low) levels of affective political polarization? The authors examine a range of possible initial causes rooted in economic and social conditions and two possible mediators -- perceived breakdown of leadership and of the social fabric (both described as types of "anomie"). Ultimately, the authors find a number of proposed causes are correlated with polarization, and also partial mediation via anomie.

While I would like to see this manuscript published in *Communications Psychology*, the manuscript requires some revision, mainly involving the provision of additional information or clarification. I also disagree with the authors on at least one important theoretical point -- I suggest they consider revising their explanation for why perceptions of affective polarization occur in response to the various causes they examine. My comments are organized below by topic.

SURVEY DATA

The authors should:

- *Provide an explanation for why the specific countries were chosen.

- *Indicate which are WEIRD and which non-WEIRD.

- *Explain why samples of N=200 were chosen.

- *Acknowledge that equal samples across countries is very problematic in one important respect: a sample of 200 represents a small country (e.g., Slovakia, with 5.4 million people) much better than a large country (e.g., China, with 1.4 billion people). This is perhaps the weakest part of the manuscript. In my view, it was a poor design choice.

- *Provide the name of the survey company/companies.

- *Explain how they recruited individuals in each country.

- *Provide some type of reassurance that recruitment worked approximately equally well in each country, for example, by indicating how close/far basic demographics in each country sample are to that country's actual demographics. Otherwise, between-country differences could be driven by the particulars of survey recruitment.

- *Provide power analyses for each country. I assume many countries are underpowered with N=200.

MEASURES/MATERIALS

My preference would be to have the entire questionnaire (in English) available in the appendix rather than an external website.

The authors should discuss how their Anomie measures are similar to other concepts/measures, such as "external efficacy" and "trust in government" (both used frequently in political science).

The authors should discuss the fact that some of their measures of societal conditions overlap conceptually with their measures of Anomie, such as Government effectiveness and Rule of law. Others do not. We would expect more mediation when the cause and mediator are conceptually related, yes? And, why not also measure things like "perceptions of economic inequality" or "perceptions of GDP"?

The authors should acknowledge the different types of measures and the implications of including all these different types. In some instances, they use "objective" measures (such as Gini coefficient). In some cases, they use what I will call "perception of experts" measures (such as Political Stability), which seem to be theorized as similar to the objective measures. In some cases, they use "individual perception" measures (such as Anomie and Perceived Climate of Polarization). There is an apples and oranges quality here. I will also note that, while Anomie has closely associated objective measures, Perceived Climate of Polarization does not. I understand the methodological barriers to doing so, but I worry this negatively affects the interpretation (more on that below).

ANALYSES

There is a danger in including so many variables in one model. This basically leads to uninterpretable models and can cause reversals in coefficients that do not reflect the bivariate relationship. (See, e.g., Achen 2005 <https://journals.sagepub.com/doi/abs/10.1080/07388940500339167>.) I suggest the authors discuss which of the associations in the multivariate models hold up in bivariate models.

Table 3 appears to show a couple of relationships that are different from what one would expect. Are these labels in Table 3 misleading? The following coefficient has an unexpected negative sign: online media fractionalization. The following has an unexpected positive association: global peace.

The authors should discuss the relative size of the statistical relationships (both direct and indirect). Ideally, they would also translate some into concrete numbers so that readers can understand the relationships in more substantive terms.

Small thing: discuss how LMMs are associated with hierarchical or multilevel models (labels some scholars are more familiar with).

THEORY/INTERPRETATION

I do not agree with the authors' assumption that perceptions of polarization are basically made up in an effort to assert some sense of personal control in a disorganized social context. Isn't the better interpretation simply that the perceptions of

political polarization reflect reality? This is also more parsimonious (Occum's Razor), reflects research in political science, and - based on my limited knowledge of international polarization - reflects political reality (see Figure 1).

Small thing: I disagree with the definition of democracy in the final discussion section, which emphasizes freedom. Complete freedom is the stuff of libertarian philosophy. Technically, democracy is the ability of residents in some geographic area to participate in their own governance. It is about "freedom" only with respect to political freedom: to participate, to express oneself, etc. It is not about other types of freedom. After all, the whole point of laws is to restrict freedom, and, in a democracy, the polity decides how much/what to restrict.

Reviewer #3 (Remarks to the Author):

Interesting study. That said, I remain unconvinced by the theoretical framing and have significant concerns about the validity of the measures, their cross-contextual equivalence, and the statistical power of the study. My comments below outline these issues in more detail:

Theoretical framing: I am not sold on the central theoretical question. Why should country-level characteristics cause perceived polarization? The authors do not provide a convincing account of why macro-level variables should be treated as causes—rather than consequences—of perceived polarization. As it stands, the causal logic feels underdeveloped.

Preregistration claim: I am unclear what the authors mean when stating, "we preregistered the aim of the study." On the OSF page, I only see two Word documents outlining survey measures—this does not constitute a preregistration by the standards laid out by Nosek and colleagues (Annual Review of Psychology, 2018). This study should not be labeled as preregistered.

A related point: the manuscript states the study was conducted in mid-2024, but it's not possible to verify whether the so-called "preregistration" occurred before, during, or after data collection.

Sample information: Conducting a study in 44 countries is impressive, but crucial sampling information is missing. Table S1 shows about 200 participants per country, but I couldn't find how they were recruited. Were these online opt-in samples? Were quotas used? What compensation was provided? This is essential methodological information and needs to be reported.

Perceived polarization measure:

Where is this measure taken from? Please provide more background on its validity.

As someone from the Netherlands, I struggle to see how this item works. The item asks respondents to reflect on the dominant groups of voters in their country. Yet the Dutch party system is extremely fragmented—so what would respondents actually think of?

This example illustrates a broader concern: I need more information on the conceptual and measurement validity of this item.

Has this measure been validated cross-nationally?

In Figure 1, the authors assume measurement equivalence and begin comparing groups. That seems premature given the concerns above.

Anomie: Has this measure been validated for cross-national use? If not, establishing measurement equivalence is essential before comparing results across contexts.

Ideological aggregation: Combining social and economic ideology is problematic. As Ariel Malka (2019, British Journal of Political Science) has shown, these dimensions function very differently across political contexts. Aggregating them may obscure rather than clarify patterns.

Figure 1: It is unclear what the error bars represent—please clarify in the figure caption or legend. The histograms can be removed; points would suffice. More generally, I would encourage a richer visualization of the underlying data. Given the cross-national nature of the study, raincloud plots or similar visualizations would make the distributions more interpretable.

Face validity: The perceived polarization measure raises further concerns. The U.S.—arguably one of the most polarized societies—scores in the mid-range, while the Netherlands, which has seen substantial elite-level polarization recently, scores lowest. I also question how meaningful a scale range of 2.5 to 3.5 is in this context.

Model overfitting: Table 3 appears to contain an overfitted model. There is a high likelihood of endogeneity among the included predictors. This should be acknowledged and addressed.

Mediation analysis: The mediation analyses are not identifiable, as discussed by Bullock & Green (JPSP). I do not think this design can support causal claims about mediation effects.

Statistical power: Although the total N is large, the country-level model includes only 44 cases. That places a heavy burden on detecting country-level effects. The authors should address whether the multilevel models are sufficiently powered to identify effects of the size and kind they are interested in.

EDITORIAL POLICIES

We ask that you ensure your manuscript complies with our editorial policies and reporting requirements.

To that end, we require revised manuscripts to be accompanied by a completed item: a reporting summary that collects information on study design and procedure.

- <https://www.nature.com/documents/nr-reporting-summary.pdf>>Nature Research Reporting Summary

Your revised manuscript can only be sent back to the referees if this checklist is completed and uploaded with the revision.

Notes: If you have submitted a Stage 1 Registered Report, Review, Primer, Comment, or Perspective you do not need to submit these forms. If you have already submitted these forms, you may disregard this request.

** Visit Nature Research's author and referees' website at <http://www.nature.com/authors>>www.nature.com/authors for information about policies, services and author benefits**

If you experience problems in linking your ORCID, please contact the <http://platformsupport.nature.com/>>Platform Support Helpdesk.

Version 1:

Decision Letter:

Dear Ms Lee,

Thank you for your patience during the peer-review process. Your manuscript titled "Exploring the perceived climate of polarization across 44 countries in relation to broader societal conditions" has now been seen by 3 reviewers, and I include their comments at the end of this message. They find your work improved, but note persistent concerns. We remain interested in the possibility of publishing your study in Communications Psychology, but would like to consider your responses to these concerns and assess a revised manuscript before we make a final decision on publication.

We therefore invite you to revise and resubmit your manuscript, along with a point-by-point response to the reviewers. Please highlight all changes in the manuscript text file.

We consider it critical that the referees' methodological and presentational concerns are fully addressed. In a similar vein, we ask that manuscripts convey a clear insight into raw data distributions, as indicated in the attached Mandatory Revision

Requests document.

While your paper is generally at a high level of compliance with our reporting policies and guidelines, a few issues remain. All requirements listed in this document need to be fully met, or the work will be returned to you for further revisions without peer review. This workflow is in place to increase the likelihood that the paper will be accepted for publication. It reduces the number of rounds of revision (and review) and ensures that the reviewers vet a version of the article that is compliant with journal policies. If you have any questions regarding the required revisions, please contact the journal prior to resubmission to avoid a negative outcome.

Please submit the following items:

- Revised manuscript
- Point-by-point response to the referees' comments
- Mandatory Revision Requests Table (attached).
- Cover letter (as a separate document)

via this link: Link Redacted .

** This url links to your confidential home page and associated information about manuscripts you may have submitted or are reviewing for us. If you wish to forward this email to co-authors, please delete the link to your homepage first **

Best regards,

Marike, on behalf of

Philipp Schmid

Philipp Schmid, PhD
External Editor
Communications Psychology

Marike Schiffer, PhD
Chief Editor
Communications Psychology

REVIEWER REPORTS:

Reviewer #1 (Remarks to the Author):

I appreciate the authors' work in revising the paper. I think it has improved. Yet, I also continue to have some concerns that I believe need to be addressed prior to publication. First, I continue to think the introduction is confusing. The authors start with affective polarization and then discuss perceived polarization. Then they introduce their measure. They never compare any of these measures (also note there is work that looks at perceptions of the other side's affective polarization – see Lees and Cikara NHB, Less et al. PNAS Nexus). I would strongly encourage the authors to substantially shorten the introduction and more succinctly make the point that they are introducing a distinct measure. It is an interesting measure, and it can stand on its own. Right now, it instead just confuses things. I also continue to find the statement about political independents misleading – in prior work, it would be only 12% that would not be included but as written it almost suggests 43% would be excluded. This is misleading. There is no need for

that. Just make the point that pure independents are typically excluded. Finally, if they want to tie affective polarization to functionality, they should see Druckman et al.'s 2024 book.

Second, the theoretical discussion that connects anomie to the measures is too abstract. More is needed to introduce the precise measures.

Third, the description of the data is incomplete. The authors state that it came from Dynata as a convenience sample and then in the appendix they report the gender distributions. Were quotas used in drawing samples? Were weights applied? If neither quotas nor weights were used I would be uncomfortable with the presentation of the country distributions. Those could easily be inaccurate due to the nature of the samples and readers may be misled in their comparative assessments of countries. I am less worried about the correlational analyses that use such samples but making points about precise estimates if the samples are indeed virtually pure convenience samples seems problematic to me.

Fourth, I do not understand why the authors do not directly add the anomie measures to Table 2 analyses. This seems like a missed opportunity. The current Table 2 is not particularly interesting. It also is confusing to say that coefficients smaller than .15 should be interpreted with caution when most of the coefficients are less than .15. It does not follow from a power analysis to ignore coefficients.

I realize the authors want to treat anomie as a mediator but that leads to my fifth point. Adding a reference to mediation problems in the limitation section is insufficient. The mediation analysis does not tell us very much as it could easily be the case that their dependent variable is shaping anomie or some other variable is. Right now, the mediation analysis is almost the centerpiece of the paper and they are not particularly credible. Instead, I would add the anomie measures as correlates in Table 2. I am fine with them including the mediation analysis but it should be stated at the outset that they are at best exploratory.

Sixth, the authors response on WEIRD countries is reasonable but then I would minimize the extent to which that is emphasized – it is odd to emphasize it and then not explore it.

In sum, I continue to find the paper interesting and an important contribution. I think addressing my concerns expressed here would not be difficult. I am afraid if the authors do not address them/revise, I cannot support publication as I worry the paper as written could be misleading (e.g., in comparing across countries, the relationship with work on affective polarization, mediation analyses). I would like to see this published and think it would be easy to do. And be a very valuable contribution. The authors may simply disagree with my concerns which is fine but it would leave me not supportive.

Reviewer #2 (Remarks to the Author):

The authors have done a nice job with their revisions. Although I still have some disagreements with respect to interpretation and also think that causal direction is awfully hard to know, I think the study is impressive in the breadth of its coverage and execution and ought to be published.

Reviewer #3 (Remarks to the Author):

I have read the revised version. While I think the authors have addressed many of my concerns. Three issues remain. I leave it up to the editor to decide what to do with them.

To be honest, I do not see convincing evidence for the validity of the used measure in terms of construct validity. Is this indeed a measure of perceived polarization. I leave it up to the editors to decide if this is an issue or not. For me, I would like to see more evidence that the outcome measure we have in this study captures the concept the authors claim to study

The problem with mediation analyses with your design is that you cannot trust the estimates as they are biased in unknown ways. See the work by Bullock & Green.

Thanks for the sensitivity analysis. If $\beta = .15$ and larger are sufficiently powered, then I think you should be careful putting any weight on smaller associations. I think the manuscript and visuals could do a better job helping the reader to draw the correct conclusions.

* **TRANSPARENT PEER REVIEW:** Communications Psychology uses a transparent peer review system. This means that we publish the editorial decision letters including Reviewers' comments to the authors and the author rebuttal letters online as a supplementary peer review file. However, on author request, confidential information and data can be removed from the published reviewer reports and rebuttal letters prior to publication. If your manuscript has been previously reviewed at

another journal, those Reviewers' comments would not form part of the published peer review file.

If you experience problems in linking your ORCID, please contact the Platform Support Helpdesk.

Version 2:

Decision Letter:

Dear Ms Lee,

Your manuscript titled "Country-level instability is related to a stronger perceived climate of polarization across 44 countries" has now been seen by our reviewers, whose comments appear below. In light of their advice I am delighted to say that we are happy, in principle, to publish a suitably revised version in Communications Psychology.

We therefore invite you to revise your paper one last time to address the remaining concerns of our reviewers and a list of editorial requests. At the same time we ask that you edit your manuscript to comply with our format requirements and to maximise the accessibility and therefore the impact of your work.

EDITORIAL REQUESTS:

SUBMISSION INFORMATION:

In order to accept your paper, we require the files listed here <https://www.nature.com/documents/commsj-file-checklist.pdf> .

OPEN ACCESS:

Link Redacted

** This url links to your confidential home page and associated information about manuscripts you may have submitted or be

reviewing for us. If you wish to forward this email to co-authors, please delete the link to your homepage first **

Best regards,

Marika Schiffer, on behalf of

Philipp Schmid, PhD
Editorial Board Member
Communications Psychology

Marika Schiffer, PhD
Chief Editor
Communications Psychology

REVIEWERS' COMMENTS:

Reviewer #1 (Remarks to the Author):

I appreciate the authors careful response to my comments. I support publications. I have a few thoughts, however.

1. The changes to the introduction are very good. I see why the authors might not want to get too engaged with the work by Lees and Cikara and Lees et al. Yet, not acknowledging them at all may give readers the impression that you are unaware of the work. If footnotes are allowed, I would simply add a footnote at the end of the second paragraph that says there is work that looks at perceived dislike by the other partisan side (Lees and Cikara, Lees et al.) You differ from this work by looking at perceptions of the system more generally.

2. The text suggests as line 345 that direct effects of anomie are entered into the model but they do not appear in Table 4 -- so I am confused. The response memo is not helpful b/c the line numbering is distinct from what I see (it likely changed when uploaded). I think somewhere there should be a multi-variate model that readers can see with those variables entered.

Overall, I look forward to seeing this in print.

Reviewer #3 (Remarks to the Author):

Thank you, I think you have addressed my comments.

14 October 2025

Resubmission – Manuscript # COMMSPSYCHOL-25-0433

Exploring the perceived climate of polarization across 44 countries in relation to broader societal conditions

Dear Dr Philipp Schmid and Reviewers,

Before turning to your comments, we would like to thank you for your constructive feedback and thoughtful engagement with our work. We have made numerous revisions to the manuscript and have endeavored to address all the concerns raised. Below, we detail the changes made in response to your comments.

Responses to Editor Comments

- 1. Thank you for your patience during the peer-review process. Your manuscript titled "Exploring the perceived climate of polarization across 44 countries in relation to broader societal conditions" has now been seen by 3 reviewers, whose comments are appended below. You will see that they find your work of some potential interest. However, they have raised quite substantial concerns that must be addressed. In light of these comments, we cannot accept the manuscript for publication, but would be interested in considering a revised version that fully addresses these serious concerns.***

We hope you will find the Reviewers' comments useful as you decide how to proceed. Should additional work allow you to address these criticisms, we would be happy to look at a substantially revised manuscript. If you choose to take up this option, please highlight all changes in the manuscript text file, and provide a detailed point-by-point reply to the reviewers.

We would like to reiterate our appreciation for the opportunity to revise and resubmit our manuscript. We have tracked our changes in the manuscript text file and provide detailed point-by-point responses to the reviewers' comments below.

- 2. Moreover, we ask that you provide a sensitivity power analysis to demonstrate the (nested) samples' power to detect the smallest effect size of theoretical significance, with a scientific justification for this target effect size. We further ask that you avoid any directional or causal claims and clarify throughout that the work is exploratory (hypotheses and analyses were not preregistered).***

We would like to thank the Editor for this feedback. In response, we conducted a power analysis to evaluate the sensitivity of our multilevel design to detect country-level effects. We now discuss the results of the power analysis in the manuscript, first under the "Method of Analysis" and then in the "Limitations and Future Directions":

- "We also conducted a post hoc sensitivity power analysis to evaluate the minimum detectable effect size given our multilevel design. Based on 44 countries with approximately 200 participants per country ($ICC = .12$), a closed-

form z-based approximation indicated that the models had 80% power ($\alpha = .05$, two-tailed) to detect standardized country-level effects of $\beta = .15$ or larger.” (Lines 334-338)

- “First, because our design had 80% power to detect standardized country-level effects of $\beta = .15$ or larger, effects smaller than this threshold should be interpreted with caution, as the study was not optimally powered to reliably detect them.” (Lines 637-639)

Additionally, we have made revisions throughout the manuscript to avoid directional or causal claims and to clarify that the research is exploratory. Previously, we acknowledged the exploratory nature of the research in the “Methods” (Lines 187-188): “However, we did not pre-register the aims and hypotheses of the current study due to its exploratory nature.” We further clarify throughout the manuscript:

- “We specifically **conducted exploratory analyses to test** the relationship between dysfunctional and disruptive country-level conditions and the perceived climate of polarization.” (Lines 170-172)
- “We sourced a variety of country-level measures from external databases to identify the macro-level societal **conditions associated with a stronger** perceived climate of polarization (see Table 1).” (Lines 296-303)
- “**Through exploratory analyses** on a new multinational dataset spanning 44 countries, we aimed to gain a clearer picture of the societal factors **associated with stronger** perceptions of a polarized societal climate.” (Lines 508-510)
- “**Although our cross-sectional mediation design precludes causal inferences, one explanation may be that** when economic hardship becomes entrenched, when violence or instability disrupt everyday life, and when the government is perceived as corrupt or ineffective, people may come to believe that society is not just struggling but unravelling.” (Lines 520-524)
- “Additionally, our cross-sectional **and exploratory** design precludes strong conclusions about the directionality of relationships.” (Lines 639-640)
- “While our mediation models explore a potential mechanism through which societal dysfunction and disruption may relate to the perceived climate of polarization, **they provide no definitive basis for causal inference.**” (Lines 641-643)
- “Drawing on data from 44 culturally diverse countries, **our exploratory analyses** found that markers of societal dysfunction and disruption were linked to stronger perceptions of a polarized societal climate.” (Lines 663-665)

3. I am attaching a checklist that details critical reporting requirements for the revised manuscript. Please attend to each item and ensure your manuscript is fully compliant. We are requesting that your manuscript aligns with these requirements as this facilitates the evaluation of your manuscript, reducing delays in re-review and potential future acceptance. If your revised manuscript is not aligned with these requests on major issues, such as those concerning statistics, it may be returned to you for further revisions without re-review. Additional information can be found in our style and formatting guide Communications Psychology formatting guide.

We thank the Editor for sharing the critical reporting requirements for the revised manuscript. We can confirm that the revised manuscript aligns with these requirements.

Responses to Reviewer 1 Comments

4. *I enjoyed this paper. There have been a lot of “negative” trends in the world such as increasing economic inequality, decreasing efficacy, decreased well-being, increased social media usage / inattentiveness. It seems intuitive that these dynamics relate to perceptions of affective polarization. I also agree that they seem more relevant to perceptions than actual affective polarization and hence the focus here makes sense. I only have a few comments (which is unusual – I typically have more to say but this strikes me as an important, very interesting, correlational demonstration).*

We wish to thank Reviewer 1 for their time reviewing our manuscript, and for their positive response. We greatly appreciate their guidance for improving our manuscript.

5. *First, the first paragraph connects affective polarization to the violation of democratic norms. This is an area of debate and at least the authors should footnote that, citing Broockman et al. 2022 (AJPS), Voelkel et al. 2023 (NHB), 2024 (Science), and Druckman et al. (book).*

We wish to thank Reviewer 1 for highlighting this oversight. We agree that there is ongoing debate about the relationship between affective polarization and democratic backsliding and appreciate the suggested references. In revising the manuscript, however, we felt that engaging with this debate in the opening paragraph risked shifting focus away from the central aims of this paper. Rather than adding a footnote, we have therefore elected not to frame affective polarization in relation to democratic backsliding. We instead refer to recent work illustrating a connection between affective polarization and the erosion of trust in democratic institutions (Lines 59-63): “Findings underscore the harmful effects of affective polarization, in particular, which **may** undermine social cohesion, **erode trust in democratic institutions**, and hinder effective policy development and governance (Iyengar et al., 2012; Iyengar et al., 2019; **Torcal & Carty, 2022**; Wilson et al., 2020; Hetherington & Rudolph, 2015).” We believe this revision allows us to highlight some of the potential negative consequences of affective polarization in a more concise manner.

6. *Second, the authors mention the exclusion of independents in many analyses. That’s a fair point although see Ahn and Mutz’s 2024 (POQ) paper. Regardless, they cite 43% of Americans as independents. I assume that this percentage is counting independent leaners as independents – and nearly all studies actually treat such respondents as partisans. Thus, I think it is likely an overstatement of how many people are excluded from these studies.*

We wish to thank Reviewer 1 for this helpful clarification. We agree that our previous framing may have overstated the extent to which political independents are excluded from research on affective polarization. As the reviewer notes, there are important examples of published work, including Ahn & Mutz (2024), that incorporate independents in analyses, and many studies classify independent leaners as partisans rather than excluding them. We have revised the manuscript to better reflect this nuance and to clarify the scope of the issue (Lines 98-108):

“In addition, reliance on feeling thermometers **may** overlook individuals who do not strongly identify with a political party. **Political independents** are often removed from

country-level analyses of affective polarization or assigned a partisan identity based on their political leaning toward a particular party (e.g., Kingzette et al., 2021; Druckman & Levendusky, 2019; Reiljan, 2020; Boxell et al., 2020). This approach risks omitting or misrepresenting an important subset of the electorate; for example, approximately 43% of American voters identified as political independents in 2023 and 12% did not lean toward either major party (Jones, 2024). Developing new measurement approaches that facilitate broader multinational data coverage and greater inclusion of non-partisan perspectives would offer a more complete picture of affective polarization around the world.”

7. *Third, the authors might tie their work to perceptions of polarization – even though much of that focuses on ideology: Levendusky and Malhotra 2016 (Poli Comm), Druckman et al. 2022 (JOP).*

We wish to thank the reviewer for this suggestion. We connect our work to existing literature on perceived polarization in the paragraph starting in Line 124. However, we agree that it would be better to clarify that existing research primarily focuses on perceived *ideological* polarization rather than perceptions of social or affective polarization. We now make this point clearer and highlight how our work builds on existing literature (Lines 124-138):

“This approach builds on growing research exploring perceived polarization, which has primarily focused on perceptions of partisan ideological divisions. Past research suggests individuals often overestimate the ideological distance and demographic differences between voter groups in society (Ahler & Sood, 2018; Levendusky & Malhotra, 2016) and these misperceptions can motivate more extreme policy positions (Ahler, 2014) and amplify dislike and distrust for political opponents (Druckman et al., 2022; Enders & Armaly, 2019). Importantly, perceptions of ideological polarization may be a stronger predictor of cross-party warmth and trust than the actual ideological distance between political parties (Enders & Armaly, 2019). This finding suggests that perceived polarization – the psychological experience of polarization – may play an important role in shaping how people think and behave within polarized contexts. Our focus on the perceived climate of polarization extends this line of work, shifting attention from perceptions of ideological divisions to the affective and relational dimensions of polarization. Attending to perceptions of antipathy and avoidance between partisan groups opens new avenues for understanding how people make sense of political polarization in their society.”

We also wish to thank the reviewer for these helpful references, which provide additional support for the idea that people tend to misperceive or overestimate the ideological differences between partisans (Levendusky & Malhotra, 2016), which can amplify animosity towards political opponents (Druckman et al., 2022). We now integrate these references to provide additional empirical support for our claims (please see the quotation above).

8. *Fourth, the authors argue that understanding perceptions of polarization provides insight into how polarization develops. I'm not sure about the nature of that connection and it seems tangential to the point of the paper. Without more evidence and logic, I would just drop that point.*

We thank Reviewer 1 for this feedback. We have now dropped this argument and instead address how our focus on the perceived climate of polarization builds on existing research related to perceptions of polarization, as suggested in comment 7. Please refer to the quotation in comment 7 for more detail.

9. *Fifth, the authors might find a recent study by Stapleton and Wolak 2024 (POQ) on efficacy and affective polarization. It is a similar conceptual argument I think.*

We wish to thank the reviewer for bringing our attention to Stapleton and Wolak's (2024) study, which we read with great interest. We agree that the study has some overlaps with our work, including a focus on the social dimensions of polarization. However, Stapleton and Wolak's primary interest was somewhat different from our own insofar as they examined the propensity to express hostility toward political opponents rather than perceptions of the broader societal climate of polarization. Additionally, their focus on individual-level predictors stood in contrast to our examination of macro-level contextual factors. Given these differences, we could not find a logical place to incorporate a citation to this work in our revised manuscript. We nonetheless wish to thank the reviewer for sharing Stapleton and Wolak's work with us.

10. *Sixth, I appreciate the care put in to the measure and the including / checking back translations. I also found the results sensible/fascinating and carefully presented. I had some concerns about causal direction, but the authors thoughtfully address this in the conclusion. The appendix is also useful.*

We thank Reviewer 1 for their positive feedback. We are delighted to hear that the reviewer appreciated these elements of the research. The reviewer's concerns about causality were echoed by the Editor and we have made revisions to avoid causal language and clarify the exploratory nature of our analyses throughout the manuscript. Please see our response to comment 2 for more information.

11. *Finally, there are three analysis questions I have:*

a) the mediation analysis is fine given norms in psychology from what I can gather but there is substantial debate in political science about this type of mediation analysis (see papers by John Bullock) – the author might acknowledge that the mediation cannot be definitive,

We wish to thank Reviewer 1 for raising this important point. We now discuss the limitations of mediation analyses in the "Limitations and Future Directions" section. Specifically, we state (Lines 641-646):

"While our mediation models explore a potential mechanism through which societal dysfunction and disruption may relate to the perceived climate of polarization, they provide no definitive basis for causal inference. As noted by Bullock et al. (2010), mediation models with unmanipulated mediators are vulnerable to biased estimates and cannot account for all possible confounds. The results of our mediation analyses should therefore be interpreted with caution. Longitudinal or experimental studies are needed to further probe causal pathways."

b) I would be interested in whether there are differences between WEIRD and non-WEIRD countries given that was an initial sampling motivation,

We appreciate the reviewer's suggestion to explore whether there are any differences between WEIRD and non-WEIRD countries. However, we are concerned that our dataset would be statistically underpowered to test interactions between WEIRD status and country-level predictors given the limited sample of 44 countries. Please see our response to the Editor (comment 2) for more detail about our sensitivity power analyses.

Additionally, we share the concerns expressed by Henrich et al. (2010), who cautioned against treating WEIRD versus non-WEIRD as a binary distinction. Such categorization risks obscuring the substantial heterogeneity within each group, while overstating the presumed differences between them. For example, Japan and South Korea are educated, industrialized, rich, and democratic but not Western. Although they would technically be categorized as "non-WEIRD", they arguably share more meaningful similarities with prototypical WEIRD societies than with a country such as Argentina, which would also be classified as "non-WEIRD" due to lower levels of education, poorer economic performance, and weaker institutions. Given the WEIRD backronym was originally intended as a mnemonic device rather than as a theoretical framework for global psychological variation, we are cautious about assuming that meaningful psychological differences align along a WEIRD to non-WEIRD continuum (Henrich et al., 2010).

c) I am not sure if this is too complicated but is there a way to look at any differences by based the government's ideology complexion (left/right)?

We thank Reviewer 1 for this suggestion. We agree that it would be interesting to investigate whether there are any interaction effects based on government ideology. However, with only 44 countries in our dataset, we are underpowered to reliably detect country-level interactions. Please see our response to the Editor (comment 2) for more detail about our sensitivity power analyses.

12. Overall, these are small comments. I really enjoyed the paper.

We thank the reviewer for their valuable feedback and are delighted to hear that they enjoyed the paper. We appreciated the opportunity to improve our work, and we hope that our revisions adequately address Reviewer 1's concerns.

Responses to Reviewer 2 Comments

13. It was a pleasure to read "Exploring the perceived climate of polarization across 44 countries in relation to broader societal conditions." In an exploratory fashion, the authors seek to answer an important question: why are countries likely to experience high (or low) levels of affective political polarization? The authors examine a range of possible initial causes rooted in economic and social conditions and two possible mediators -- perceived breakdown of leadership and of the social fabric (both described as types of "anomie"). Ultimately, the authors find a number of proposed causes are correlated with polarization, and also partial mediation via anomie.

While I would like to see this manuscript published in Communications Psychology, the manuscript requires some revision, mainly involving the provision of additional information or clarification. I also disagree with the authors on at least one important theoretical point -- I suggest they consider revising their explanation for why

perceptions of affective polarization occur in response to the various causes they examine. My comments are organized below by topic.

We wish to thank Reviewer 2 for their detailed feedback and for the time taken to review our manuscript. We greatly appreciate their comments, which provided opportunities for us to substantially improve our manuscript. We address each of their points below.

14. SURVEY DATA

The authors should:

a. *Provide an explanation for why the specific countries were chosen.*

We thank Reviewer 2 for raising this important omission. The specific countries were selected based on the availability of survey panels through Dynata. As a result, our sample should be considered a convenience sample. We now clarify this in the manuscript (Lines 219-221): “The 44 countries were selected based on availability through Dynata. As such, the sample represents a convenience sample.”

b. *Indicate which are WEIRD and which non-WEIRD.*

We wish to thank the reviewer for this suggestion, which is closely related to the feedback provided by Reviewer 1 in comment 11b. Please refer to our response to Reviewer 1, in which we outline our concerns about this approach.

c. *Explain why samples of N=200 were chosen.*

We appreciate the reviewer’s request for clarification about our sampling rationale. The data for this study were collected as part of a larger research project designed to test multiple hypotheses across different studies. We determined the sample size of $N = 200$ per country based on previous multinational datasets (e.g., Kirkland et al., 2022; Sprong et al., 2019) and the cost of recruiting participants through Dynata.

These details are available in our pre-registration for the data collection method on the Open Science Framework. We now also provide greater clarity about this design choice in Lines 212-214 of the manuscript: “Based on the sample sizes obtained in past multinational datasets (e.g., Kirkland et al., 2022a; Sprong et al., 2019) and the cost of recruiting participants through Dynata, we aimed to recruit a minimum of 200 participants from 44 countries ($N = 8917$) [...]”.

d. *Acknowledge that equal samples across countries is very problematic in one important respect: a sample of 200 represents a small country (e.g., Slovakia, with 5.4 million people) much better than a large country (e.g., China, with 1.4 billion people). This is perhaps the weakest part of the manuscript. In my view, it was a poor design choice.*

We thank Reviewer 2 for raising this concern. Our design follows the standard approach in comparative survey research (e.g., European Social Survey, World Values Survey, International Social Survey Programme), where each country contributes a fixed or capped sample rather than a population-proportional sample. The reason is that our analytic focus is at the country level – treating each country as one data point – and thus we require sufficient within-country respondents to stabilize national means, rather than population-representative estimates of each country’s distribution. Using equal samples

prevents countries with very large populations from dominating the analysis, which is particularly important in multilevel modeling where Level-2 predictors operate at the country level. While we acknowledge that fixed N's do not permit population-weighted prevalence estimates, this was not our study's aim, and we view our design as consistent with best practice in cross-national research.

However, we acknowledge that the recruitment of non-representative samples limits the generalizability of our findings. We now explain this limitation in the penultimate paragraph of the paper (Lines 651-653): "Finally, our country samples were not representative of population demographics, limiting the generalizability of findings."

e. Provide the name of the survey company/companies.

We wish to thank Reviewer 2 for highlighting this oversight. We have now clarified the survey company in Line 211: "Data were collected by an external survey company, Dynata, between May and June 2024."

f. Explain how they recruited individuals in each country.

We thank the reviewer for this pointing out this oversight. We now explain how Dynata recruited participants from each country (Lines 221-223): "Dynata recruited participants through pre-established panels in each country. All participants provided informed consent and were compensated monetarily for their participation."

g. Provide some type of reassurance that recruitment worked approximately equally well in each country, for example, by indicating how close/far basic demographics in each country sample are to that countries actual demographics. Otherwise, between-country differences could be driven by the particulars of survey recruitment.

We thank Reviewer 2 for this suggestion. We agree that assessing whether country samples align with population-level demographics is important for evaluating the robustness of cross-national comparisons. However, our ability to conduct these checks was limited by a mismatch between our demographic measures and the types of national statistics that are available across the 44 countries we sampled. For example, our study only included adult participants, precluding comparisons with population age distributions that typically include minors. Additionally, our measure of subjective socioeconomic status captured *relative perceptions* of social standing rather than objective income levels. Furthermore, our religiosity measure assessed the importance of religion in daily life, which cannot be compared against census data on religious affiliation or membership rates. Finally, our measures of political orientation asked participants to locate themselves on a left-right continuum, rather than their party membership or vote choice.

As a result, gender was the only variable for which meaningful population-level comparisons were possible. We now report these in Supplementary Materials 2. Across the 44 samples, gender ratios generally aligned with population benchmarks (mean deviation = 1.4 percentage points, range = 0-16.4). Only two samples exceeded ± 10 points: Slovakia and the United Arab Emirates. However, it is important to note that we control for demographic measures – including gender – in all models reported in the manuscript. We also report estimates without demographic controls in the Supplementary

Materials, and the pattern of results does not change when these variables are excluded from analyses.

We acknowledge that these limitations restrict the degree of reassurance we can provide about the representativeness of our samples. We remain open to further suggestions from the reviewer in future revisions.

h. Provide power analyses for each country. I assume many countries are underpowered with $N=200$.

We wish to thank the reviewer for sharing their concerns about statistical power. While each country sample included approximately 200 participants, our use of linear mixed models (i.e., hierarchical or multilevel models) did not rely on per-country estimates. Rather, statistical power for individual-level predictors derived from the total sample across countries ($N = 8917$), while power for country-level predictors was determined by the number of countries ($N = 44$). We therefore decided it would not be appropriate to conduct power analyses for each country individually.

However, in response to concerns expressed by the Editor (comment 2) and Reviewer 3 (comment 30), we conducted a power analysis to determine whether our linear mixed models were sufficiently powered to identify country-level effects. Please refer to our response to comment 2 for more information.

15. MEASURES/MATERIALS

a. My preference would be to have the entire questionnaire (in English) available in the appendix rather than an external website.

We thank Reviewer 2 for this suggestion. In response, we now direct readers to the Supplementary Materials, where we have included all survey items relevant to the current study (Line 242): “See Supplementary Materials 3 for the exact wording of all survey items”

We decided not to include the full questionnaire because many of the items pertain to data that were not used in the current study. We believe that including these items could create unnecessary confusion for readers. However, the full survey is publicly available in our pre-registration for the data collection method and interested readers can easily access the full survey via the link provided (Lines 184-187): “The data for this study came from a larger survey involving measures that are not discussed here. On March 27, 2024, we pre-registered the data collection method and measures for the larger survey on the Open Science Framework (OSF; <https://osf.io/ghjdu/overview>).”

b. The authors should discuss how their Anomie measures are similar to other concepts/measures, such as "external efficacy" and "trust in government" (both used frequently in political science).

We appreciate the reviewer’s suggestion to situate the anomie measure in relation to constructs such as external efficacy and trust in government, which are more frequently used in political science. While both external efficacy and trust in government capture perceptions of institutional responsiveness and legitimacy, Teymoori et al.’s (2016)

measure of anomie is designed to reflect a broader collective sense of societal breakdown. It encompasses two dimensions: (1) *disintegration*, the perception that the social fabric is decaying, including a loss of shared trust, moral standards, and social cohesion; and (2) *dysregulation*, the perception that leadership is unfair, illegitimate, or ineffective. Anomie arises when there is a shared sense that the two pillars of a functioning society – strong social cohesion and effective leadership – are eroding.

In this way, anomie is conceptually related to, but broader than, constructs such as external efficacy and trust in government. Whereas the latter focus on more specific perceptions of political institutions and processes, anomie encompasses wider perceptions of societal erosion across both governance and the social fabric. We therefore view anomie as providing a more comprehensive framework for understanding how individuals evaluate the overall state of their society.

In the manuscript, we now situate the anomie measure in relation to more frequently used constructs (Lines 278-283): “Both subscales were included to capture broader perceptions of societal decay. Although related measures such as external efficacy and trust in government may be more common in the literature, these emphasize narrower judgments about institutional responsiveness and legitimacy. In contrast, the Perceptions of Anomie scale encompasses evaluations of not only political institutions and processes, but also the wider social context in which they are embedded.”

- c. The authors should discuss the fact that some of their measures of societal conditions overlap conceptually with their measures of Anomie, such as Government effectiveness and Rule of law. Others do not. We would expect more mediation when the cause and mediator are conceptually related, yes? And, why not also measure things like "perceptions of economic inequality" or "perceptions of GDP"?***

We wish to thank Reviewer 2 for raising this important point. By including Teymoori et al.'s (2016) measure of anomie as a mediator, our goal was to examine one pathway through which dysfunctional and disruptive societal conditions may relate to perceptions of a polarized societal climate. Specifically, we sought to test whether adverse societal conditions might foster more general perceptions that society is in decay, which in turn may shape how individuals view relations between opposing voter groups. Our focus, therefore, was not on perceptions of specific issues, but broader evaluations of society that could serve as a bridge between external conditions and perceptions of polarization.

We acknowledge that the Perceived Breakdown of Leadership subscale shares some conceptual ground with indicators such as Government Effectiveness and Rule of Law. However, while the former captures mass perceptions of government functioning, the latter are composite indicators based on both “objective” and expert assessment data from multiple sources (see World Bank documentation for more information: <https://www.worldbank.org/en/publication/worldwide-governance-indicators/documentation#1>). As explained in our response to comment 17a below, it is important to recognize that perceptions are not always veridical. While we agree that mediation seems more likely to emerge where there is greater conceptual overlap between

measures, we found larger indirect effects through the Perceived Breakdown of Social Fabric subscale, even for governance-related indicators. This pattern seems to suggest that individuals' perceptions of society may not map neatly onto "objective" or expert assessments of societal conditions, and that the perceived erosion of the social fabric may be particularly relevant to how people interpret the broader climate of polarization.

We appreciate the reviewer's suggestion to examine perceptions of specific issues, such as economic inequality or GDP. While this was not the approach we took in the present study, we view it as an interesting direction for future research that could complement our more general framework.

- d. The authors should acknowledge the different types of measures and the implications of including all these different types. In some instances, they use "objective" measures (such as Gini coefficient). In some cases, they use what I will call "perception of experts" measures (such as Political Stability), which seem to be theorized as similar to the objective measures. In some cases, they use "individual perception" measures (such as Anomie and Perceived Climate of Polarization). There is an apples and oranges quality here. I will also note that, while Anomie has closely associated objective measures, Perceived Climate of Polarization does not. I understand the methodological barriers to doing so, but I worry this negatively affects the interpretation (more on that below).*

We thank the reviewer for raising this point. As the reviewer notes, our study combines different types of measures, including "objective" indicators (e.g., Gini coefficient), expert assessments (e.g., political stability), and individual perceptions (e.g., perceptions of anomie and the perceived climate of polarization). This approach was necessary because our central aim was to evaluate how broad societal conditions relate to individual perceptions of the societal climate. Complementing "objective" measures with expert assessments allowed us to cover a greater variety of conditions that reflect societal dysfunction and disruption. At the same time, including perceptions of anomie offered a theoretically meaningful bridge that may help to explain how distal societal factors could be interpreted and translated into perceptions of a polarized societal climate.

This type of research question and methodological approach is not uncommon in psychology (e.g., Kirkland et al., 2024; Sprong et al., 2019). Rather than a limitation, we view the inclusion of different types of measures as an important strength in our research design. Demonstrating consistent relationships across measures derived from different origins suggests that the patterns observed are not simply artifacts of measurement. This provides more convincing support for the relationship between societal dysfunction and disruption and the perceived climate of polarization than if all measures were solely based on individual perceptions.

In the manuscript, we now acknowledge that the country-level measures analyzed derive from different sources and provide justification for this choice. Specifically, we state (Lines 305-307): "We included both "objective" indicators (e.g., GDP PPP per capita) and expert assessments (e.g., political instability) to explore a greater variety of societal conditions in the analyses."

16. ANALYSES

- a. *There is a danger in including so many variables in one model. This basically leads to uninterpretable models and can cause reversals in coefficients that do not reflect the bivariate relationship. (See, e.g., Achen 2005 <https://journals.sagepub.com/doi/abs/10.1080/07388940500339167>.) I suggest the authors discuss which of the associations in the multivariate models hold up in bivariate models.*

We wish to thank Reviewer 2 for highlighting these concerns. We would like to clarify that all country-level variables were inputted in separate linear mixed models rather than in a single multivariate linear mixed model. We explain our method of analysis in Lines 316-320: “We then centered and scaled all country-level measures and assessed their effects in separate LMMs that controlled for the demographic measures. For each model, the perceived climate of polarization was programmed as the outcome variable, the country-level and demographic measures as fixed effects, and country as a random intercept.”

In the manuscript, we also refer readers to Supplementary Materials 5, where we provide estimates for models without demographic variables. These models explore the bivariate relationship between each country-level variable and the perceived climate of polarization. The results were largely consistent with the multivariate models that included demographic variables.

- b. *Table 3 appears to show a couple of relationships that are different from what one would expect. Are these labels in Table 3 misleading? The following coefficient has an unexpected negative sign: online media fractionalization. The following has an unexpected positive association: global peace.*

We thank the reviewer for this feedback. We have now revised the labels for “online media fractionalization” and “global peace” to avoid confusion. Specifically, we have replaced “online media fractionalization” with “online media consistency”. We have also relabeled “global peace” as “conflict and instability”.

- c. *The authors should discuss the relative size of the statistical relationships (both direct and indirect). Ideally, they would also translate some into concrete numbers so that readers can understand the relationships in more substantive terms.*

We thank the reviewer for this suggestion. We agree that discussing the relative size of statistical relationships would help readers better interpret our findings. Because predictors in all models were standardized ($M = 0$, $SD = 1$), coefficients are directly comparable and can be interpreted as the expected change in the perceived climate of polarization (in raw units) for one standard deviation increase in each predictor. We explain this in the notes below the relevant tables of results. We now also provide substantive translations of selected effects to help readers interpret their practical significance:

- “The LMMs featuring country-level indicators found a stronger perceived climate of polarization in countries with less favorable economic conditions (lower GDP PPP per capita; greater income inequality, unemployment, and youth NEET), as well as heightened violence and conflict (greater conflict and instability, political

violence, and homicide). Governance-related indicators were the strongest correlates: a one standard deviation increase in government effectiveness corresponded to a 0.20-point lower score on the perceived climate of polarization, with similar associations observed for corruption control (-0.18) and rule of law (-0.17). These relationships were comparable in size to those for youth NEET and conflict and instability, for which a one standard deviation increase was associated with a ~0.15-point higher score on the perceived climate of polarization.

Relatively weaker, but statistically significant, associations were observed for GDP PPP per capita, income inequality, homicide, and political violence.

Indicators of the digital media landscape were also related: lower consistency in online media and higher internet and social media use were each associated with a stronger perceived climate of polarization. By contrast, environmental and public health indicators (droughts, floods, and extreme temperatures; pathogen prevalence; food insecurity; childhood mortality; and life expectancy) and democratic strength showed non-significant and near-zero relationships (see Table 3).” (Lines 381-413)

- “Parallel mediation analyses showed that the significant associations between country-level measures and the perceived climate of polarization were partly explained by perceptions of breakdown in the social fabric and, to a lesser extent, the leadership (see Table 4). For several indicators – including GDP PPP per capita, conflict and instability, political stability, rule of law, daily internet use, and daily social media use – only perceptions of social fabric breakdown partially explained shared variance with perceived polarization. For example, a one standard deviation decrease in GDP PPP per capita was associated with a 0.13-point higher perceived polarization score, of which 23.1% was statistically attributable to perceptions of the social fabric. Meanwhile, the indirect pathway through perceived leadership breakdown was near-zero and non-significant. Across these country-level indicators, a perceived breakdown in the social fabric explained 27.5-30.8% of the overall association with the perceived climate of polarization.” (Lines 439-450)
- “For other indicators, perceptions of both social fabric and leadership breakdown partially explained associations with the perceived climate of polarization. Income inequality, unemployment, youth NEET, homicide, political violence, government effectiveness, corruption control, and online media consistency had significant indirect pathways through both mediators, though the contribution of social fabric perceptions was consistently larger. For example, 8.3% of the association between income inequality and the perceived climate of polarization could be explained via perceptions of leadership breakdown, compared to 33.3% via perceptions of social fabric breakdown. Similarly, a perceived breakdown in the social fabric accounted for 30.8% of the association between homicide and perceived polarization, while perceived leadership breakdown only accounted for 7.7%.” (Lines 451-460)

d. *Small thing: discuss how LMMs are associated with hierarchical or multilevel models (labels some scholars are more familiar with).*

We thank Reviewer 2 for suggesting this clarification point. We now make clear the links between linear mixed models, hierarchical models, and multilevel models under the “Method of Analysis” (Lines 312-315): “Data from the current study were collected from 44 samples. To account for the nesting of individuals (Level 1) within countries (Level 2),

we conducted a series of linear mixed models (LMMs) – also referred to as hierarchical or multilevel models – using Rstudio with the lme4 package (Bates et al., 2015).”

17. THEORY/INTERPRETATION

- a. *I do not agree with the authors' assumption that perceptions of polarization are basically made up in an effort to assert some sense of personal control in a disorganized social context. Isn't the better interpretation simply that the perceptions of political polarization reflect reality? This is also more parsimonious (Occum's Razor), reflects research in political science, and - based on my limited knowledge of international polarization - reflects political reality (see Figure 1).***

We thank the reviewer for this feedback. We agree that perceptions of the climate of polarization are likely to reflect political reality to some extent. Indeed, our cross-national rankings broadly align with prior multinational research on affective polarization (e.g., Reiljan, 2020; Gidron, Adams, & Horne, 2020). At the same time, extensive research in psychology demonstrates that people’s perceptions of reality are not fully veridical and may depart from objective conditions in ways that serve psychological needs, such as the need for coherence, control, and certainty. For example, work on the Just World Hypothesis (Lerner, 1980) shows that people interpret situations in ways that preserve a sense of order, justice, and predictability, even when such interpretations distort reality. Similarly, research on System Justification Theory (Jost & Banaji, 1994), Conspiracy Beliefs (e.g., Douglas, Sutton, & Cichocka, 2017), the False Consensus Effect (e.g., Mullen et al., 1985), and Cognitive Dissonance Theory (Festinger, 1957) document the ways in which perceptions are often motivated or biased, rather than clear-eyed.

We do not have direct evidence to establish whether perceptions of the climate of polarization accurately map onto objective conditions. However, existing research on false polarization suggests people tend to misperceive the demographic composition and ideological distance between partisan groups (e.g., Ahler & Sood, 2018; Levendusky & Malhotra, 2016). Given these findings and the extensive literature demonstrating that perceptions of reality are often shaped by psychological needs, we believe it is reasonable to consider the possibility that the perceived climate of polarization may likewise be motivated, at least in part, by such processes.

- b. *Small thing: I disagree with the definition of democracy in the final discussion section, which emphasizes freedom. Complete freedom is the stuff of libertarian philosophy. Technically, democracy is the ability of residents in some geographic area to participate in their own governance. It is about "freedom" only with respect to political freedom: to participate, to express oneself, etc. It is not about other types of freedom. After all, the whole point of laws is to restrict freedom, and, in a democracy, the polity decides how much/what to restrict.***

We wish to thank Reviewer 2 for this feedback. We agree that our previous definition of democratic strength lacked specificity. Our definition was based on the one provided by the Economist Intelligence Unit, whose index was used in our analyses (2024; p. 63): “Democracy can be seen as a set of practices and principles that institutionalise, and thereby, ultimately, protect freedom.” We no longer use this wording as the paragraph that included this text has been removed from the manuscript to avoid interpretation of null results.

Responses to Reviewer 3 Comments

18. Interesting study. That said, I remain unconvinced by the theoretical framing and have significant concerns about the validity of the measures, their cross-contextual equivalence, and the statistical power of the study. My comments below outline these issues in more detail.

We wish to thank Reviewer 3 for their constructive comments and for their time reviewing our paper. Their feedback provided a valuable opportunity to improve our manuscript, and we hope that our revisions adequately address their concerns.

19. Theoretical framing: I am not sold on the central theoretical question. Why should country-level characteristics cause perceived polarization? The authors do not provide a convincing account of why macro-level variables should be treated as causes—rather than consequences—of perceived polarization. As it stands, the causal logic feels underdeveloped.

We thank the reviewer for raising this concern. Given the cross-sectional nature of our study, we agree that it would be inappropriate to make strong causal claims about the effect of country-level characteristics on the perceived climate of polarization. Consistent with requests by the Editor (comment 2) and Reviewer 1 (comment 11), we have now revised our use of causal language throughout the manuscript.

Rather than asserting causality, we theorize that conditions of societal dysfunction and disruption may contribute to a felt sense of instability. Individuals may respond to such conditions by drawing sharper boundaries between political camps as a way of managing uncertainty or reasserting coherence. This perspective is grounded in extant psychological research showing that perceptions of social reality are not simply veridical reflections of external conditions but are filtered through motivational processes, such as the need to maintain certainty, coherence, and control (see also our response to Reviewer 2, comment 17a).

Moreover, many of our predictors seem unlikely to be the consequences of the perceived climate of polarization. For example, it seems implausible that individual perceptions of polarization could drive GDP PPP per capita, unemployment rates, or levels of social media penetration. In such cases, it seems more reasonable to interpret these indicators as influences on the sociopolitical contexts within which individuals form judgments about their societies.

Thus, while we refrain from making strict causal claims, we propose that examining conditions of societal dysfunction and disruption may offer a promising lens for understanding cross-national differences in the perceived climate of polarization.

20. Preregistration claim: I am unclear what the authors mean when stating, "we preregistered the aim of the study." On the OSF page, I only see two Word documents outlining survey measures—this does not constitute a preregistration by the standards laid out by Nosek and colleagues (Annual Review of Psychology, 2018). This study should not be labeled as preregistered.

We wish to thank Reviewer 3 for this feedback. This study involved data that was collected as part of a larger research project. We pre-registered the method of data collection for the broader research project in March 2024. Unfortunately, the link included in the original manuscript did not provide direct access to the pre-registration. We apologize for this oversight. We have now corrected that error and provide a direct link to the pre-registration in Lines 185-187: “On March 27, 2024, we pre-registered the data collection method and measures for the larger survey on the Open Science Framework (OSF; <https://osf.io/ghjdu/overview>).”

Due to the exploratory nature of this particular study, we did not pre-register our aims and hypotheses. We make this point explicit in the “Methods” section, where we state (Lines 187-188): “However, we did not pre-register the aims and hypotheses of the current study due to its exploratory nature.”

21. *A related point: the manuscript states the study was conducted in mid-2024, but it’s not possible to verify whether the so-called “preregistration” occurred before, during, or after data collection.*

We thank the reviewer for highlighting this point. The updated link now provides direct access to the pre-registration for our data collection method for the broader research project. The pre-registration was time-stamped on March 27, 2024. We have also revised the manuscript to clarify the dates of data collection (Line 211): “Data were collected by an external survey company, Dynata, between May and June 2024.”

22. *Sample information: Conducting a study in 44 countries is impressive, but crucial sampling information is missing. Table S1 shows about 200 participants per country, but I couldn’t find how they were recruited. Were these online opt-in samples? Were quotas used? What compensation was provided? This is essential methodological information and needs to be reported.*

We wish to thank the reviewer for calling our attention to this important oversight, which was also raised by Reviewer 2 in comment 14. We agree that more information should have been provided about participant recruitment, and we have now done so in the revised manuscript. Please see our response to comment 14 for more detail.

23. *Perceived polarization measure:*

a. *Where is this measure taken from? Please provide more background on its validity.*

We thank the reviewer for raising this point. We developed our measure of the perceived climate of polarization by drawing on features of affective polarization that are commonly discussed in the literature. Past research specifically highlights three social trends that characterize affective polarization: (1) dislike, (2) distrust, and (3) social distance or avoidance between opposing partisans (e.g., Iyengar et al., 2012; Iyengar et al., 2019; Kingzette et al., 2021). Widely used measures of affective polarization also aim to capture these three dimensions. While trait-based ratings (e.g., seeing the outgroup as cold, incompetent, or dishonest) are also commonly used (Druckman & Levendusky, 2019), we judged them to be less suitable for representing perceptions of the overall societal climate of polarization. We believed it would be challenging to ask respondents – particularly non-partisans – to assess the degree to which voter groups make trait-based judgments of

one another. For this reason, we focused the wording of our items on perceptions of the levels of dislike, distrust, and social distance between voter groups.

In the manuscript, we now clarify the theoretical rationale for the measure (Lines 246-251): “These items reflect three social trends – dislike, distrust, and social distance or avoidance – that are commonly discussed as indicators of affective polarization (e.g., Iyengar et al., 2012; Iyengar et al., 2019; Druckman & Levendusky, 2019; Kingzette et al., 2021). By focusing on these salient dimensions of affective polarization, we aimed to assess perceptions of polarization in the social sphere, beyond ideological differences between major voter groups.”

- b. As someone from the Netherlands, I struggle to see how this item works. The item asks respondents to reflect on the dominant groups of voters in their country. Yet the Dutch party system is extremely fragmented—so what would respondents actually think of?***

We wish to thank the reviewer for sharing their concerns about the applicability of our measure across different political systems, including fragmented multi-party systems. Our scale was intentionally worded to capture respondents’ perceptions of polarization rather than impose a specific definition of the dominant groups of voters in each country. Our use of general language was necessary to ensure that the measure was applicable across different contexts. Moreover, we wanted participants to interpret the question in a way that reflects the most salient cleavages in their national context, which may not necessarily be the divisions between specific political parties. For example, in fragmented political systems such as the Netherlands, respondents may think of broad, opposing ideological camps (e.g., progressive vs. conservative), political coalitions, or other groups that they perceive as politically or socially influential.

- c. This example illustrates a broader concern: I need more information on the conceptual and measurement validity of this item.***

Has this measure been validated cross-nationally?

In Figure 1, the authors assume measurement equivalence and begin comparing groups. That seems premature given the concerns above.

We thank Reviewer 3 for this feedback. To address potential concerns about comparability, we have conducted analyses of measurement invariance for the scale across countries. We present the results of these analyses in Supplementary Materials 4:

We tested measurement invariance in three steps. First, we fitted a configural model to the data – that is, a multi-group confirmatory factor analysis (MGCFA) model without restrictions on parameters. As shown in the table below, this model demonstrated excellent fit, indicating that the overall factor structure was comparable across countries. Next, we tested a metric invariance model, which constrained factor loadings across groups. The Comparative Fit Index (CFI) decreased by .006 but fell within the recommended threshold of .01 (Chen, 2007). Although changes in the Root Mean Square Error of Approximation (RMSEA) and Standardized Root Mean Squared Residual (SRMR) exceeded conventional thresholds, these indices are known to be sensitive to model complexity, which can disadvantage more parsimonious models (Putnick & Bornstein, 2017). We therefore concluded that the data provided sufficient support for metric invariance. Finally, we tested a scalar invariance model, which constrained both

factor loadings and intercepts across countries. This model did not satisfy conventional fit criteria (Chen, 2007) and thus scalar invariance was not supported.

	CFI scaled	RMSEA scaled	SRMR	Chi- square	Δ CFI scaled	Δ RMSEA scaled	Δ SRMR
Configural	1.000	0	0	0			
Metric	0.994	0.057	0.039	152.690	-0.006	0.057	0.039
Scalar	0.976	0.084	0.055	427.839	-0.018	0.027	0.016

It is worth noting that there is growing debate about the necessity of measurement invariance testing in psychology, particularly in cross-national research. As Kusano et al. (2025) argue, strict adherence to conventional methods for measurement invariance testing may impose unrealistic expectations about how social and political constructs vary across societies. Kusano and colleagues show how measurement non-invariance may reflect genuine societal and cultural differences rather than sample bias and therefore emphasize the importance of evaluating construct validity in context.

In light of this ongoing debate, we believe that the perceived climate of polarization can be meaningfully compared across countries, despite the mixed results of our measurement invariance analyses. However, we welcome suggestions from the reviewer for alternative methods that may be used to further test measurement invariance of the scale across countries.

24. *Anomie: Has this measure been validated for cross-national use? If not, establishing measurement equivalence is essential before comparing results across contexts.*

We thank the reviewer for raising this concern. Teymoori et al.'s (2016) Perception of Anomie scale was validated for cross-national use by Kirkland et al. (2022) in Social Psychological and Personality Science. In their study featuring 37 countries, the authors found that partial metric invariance was supported by the data, as reported in their Supplementary Materials. We refer the reviewer to page 21 of their Supplementary Materials for more detail.

25. *Ideological aggregation: Combining social and economic ideology is problematic. As Ariel Malka (2019, British Journal of Political Science) has shown, these dimensions function very differently across political contexts. Aggregating them may obscure rather than clarify patterns.*

We appreciate the reviewer's concern about aggregating economic and social political orientation. We have now rerun all analyses featuring demographic variables without aggregating political orientation across these dimensions. The new analyses produced estimates that were largely consistent with prior results. However, we have revised the manuscript to reflect any inconsistencies. We thank the reviewer for this feedback.

26. *Figure 1: It is unclear what the error bars represent—please clarify in the figure caption or legend. The histograms can be removed; points would suffice. More generally, I would encourage a richer visualization of the underlying data. Given the cross-national nature of the study, raincloud plots or similar visualizations would make the distributions more interpretable.*

We wish to thank the reviewer for this insightful suggestion. We agree that raincloud plots would provide a richer visualization of the underlying data. As a result, we have now removed the histograms and instead present the distribution of the perceived climate of polarization scores for each country in raincloud plots. These are plotted alongside boxplots that indicate the interquartile range, median, and minimum and maximum scores for each country. We also indicate country mean scores in the figure. We believe these revisions have greatly enhanced data visualization.

27. Face validity: The perceived polarization measure raises further concerns. The U.S.—arguably one of the most polarized societies—scores in the mid-range, while the Netherlands, which has seen substantial elite-level polarization recently, scores lowest. I also question how meaningful a scale range of 2.5 to 3.5 is in this context.

We thank Reviewer 3 for sharing these concerns and appreciate the importance of ensuring the face validity of our measure. Recent research has challenged the notion that the United States is the most affectively polarized country. For example, Reiljan (2020) found that several European countries, including Spain, France, Slovakia, and Poland, displayed higher levels of affective polarization than the United States, whereas the Netherlands showed relatively lower levels. Similarly, in their comparative analysis of affective polarization in the United States and 19 other Western democracies, Gidron, Adams, and Horne (2020) found that American affective polarization is not extreme from a comparative perspective. Their findings also highlight the Netherlands as a low-polarization context.

We believe that the consistency between our findings and prior research supports the face validity of our measure for the perceived climate of polarization. We also note that our measure is explicitly designed to capture mass polarization rather than elite-level polarization and therefore uses wording that refers to “dominant voter groups” rather than political leaders or institutions.

However, if there is additional empirical evidence that the reviewer would like us to consider, we are happy to integrate this work into our discussion in a future round of revisions.

28. Model overfitting: Table 3 appears to contain an overfitted model. There is a high likelihood of endogeneity among the included predictors. This should be acknowledged and addressed.

We wish to thank the reviewer for highlighting these concerns. We would like to clarify that each row of Table 3 represents the output of a separate linear mixed model. In other words, the predictors were not inputted together in a single linear mixed model. We explain our method of analysis in the “Methods” section of the paper. Specifically, we state in Lines 316-320: “We then centered and scaled all country-level measures and assessed their effects in separate LMMs that controlled for the demographic measures. For each model, the perceived climate of polarization was programmed as the outcome variable, the country-level and demographic measures as fixed effects, and country as a random intercept.” In the note under Table 3 (Line 434), we further clarify: “Each line indicates a separate LMM.”

We agree that a high likelihood of interrelatedness between the country-level predictors should be acknowledged. We discuss this limitation in the penultimate paragraph of the manuscript (Lines 647-651): “Furthermore, while we aimed to explore a variety of potential country-level correlates of the perceived climate of polarization, our analyses were non-exhaustive and many of the country-level variables examined are likely interrelated. Future research should explore the role of other societal conditions, as well as how different factors may interact to shape perceptions of hostility and aversion between political groups.”

We hope these clarification points adequately address Reviewer 3’s concerns about the data presented in Table 3.

29. *Mediation analysis: The mediation analyses are not identifiable, as discussed by Bullock & Green (JPSP). I do not think this design can support causal claims about mediation effects.*

We wish to thank the reviewer for this feedback, which echoes concerns expressed by the Editor (comment 2) and Reviewer 1 (comment 11). We now address these concerns and have revised our use of causal language throughout the manuscript. Please see our response to comment 2 and comment 11 for more detail.

30. *Statistical power: Although the total N is large, the country-level model includes only 44 cases. That places a heavy burden on detecting country-level effects. The authors should address whether the multilevel models are sufficiently powered to identify effects of the size and kind they are interested in.*

We thank the reviewer for raising this concern, which was echoed by the Editor (comment 2) and Reviewer 2 (comment 14). We have now conducted a power analysis to determine whether our models were sufficiently powered to detect country-level effects. Please refer to our response to comment 2 for more detail.

6 January 2026

Resubmission – Manuscript # COMMSPSYCHOL-25-0433

Country-level instability is related to a stronger perceived climate of polarization across 44 countries

Dear Dr Philipp Schmid and Reviewers,

We would like to reiterate our appreciation for your constructive comments and engagement with our work. In response to your feedback, we have made additional revisions to the manuscript and have tried our best to address the issues that were raised. Please see our responses to your comments below. Please note that all references to line numbers pertain to the manuscript document with tracked changes enabled.

Responses to Editor Comments

- 1. Thank you for your patience during the peer-review process. Your manuscript titled "Exploring the perceived climate of polarization across 44 countries in relation to broader societal conditions" has now been seen by 3 reviewers, and I include their comments at the end of this message. They find your work improved, but note persistent concerns. We remain interested in the possibility of publishing your study in Communications Psychology, but would like to consider your responses to these concerns and assess a revised manuscript before we make a final decision on publication.***

We therefore invite you to revise and resubmit your manuscript, along with a point-by-point response to the reviewers. Please highlight all changes in the manuscript text file.

We consider it critical that the referees' methodological and presentational concerns are fully addressed. In a similar vein, we ask that manuscripts convey a clear insight into raw data distributions, as indicated in the attached Mandatory Revision Requests document.

We would like to thank the Editor for the opportunity to revise and resubmit our manuscript. We have tracked our changes in the updated manuscript file and respond to each of the reviewers' points below. As requested, we also provide greater clarity about the raw data distributions for demographic variables. Please refer to Table 2 in the updated manuscript and to Supplementary Materials 1 for a breakdown of sample details by country.

- 2. While your paper is generally at a high level of compliance with our reporting policies and guidelines, a few issues remain. All requirements listed in this document need to be fully met, or the work will be returned to you for further revisions without peer review. This workflow is in place to increase the likelihood that the paper will be accepted for publication. It reduces the number of rounds of revision (and review) and ensures that the reviewers vet a version of the article that is compliant with journal policies. If you***

have any questions regarding the required revisions, please contact the journal prior to resubmission to avoid a negative outcome.

We have endeavored to address all additional feedback provided in the Editorial Request Table. Please see our tracked changes in the revised manuscript file for more detail. We have specifically made the following revisions:

1. We have revised the manuscript title so that it now conveys a result.
2. We have ensured that the Abstract is included in the manuscript on page 2.
3. We have removed language such as “new”, “novel”, etc. throughout the manuscript.
4. We have clarified all abbreviations when they are first mentioned.
5. We have ensured consistent use of “Türkiye” when referring to the country.
6. We have ensured that the Introduction does not include a summary of results.
7. We provide a full overview of demographics across the pooled sample (see Table 2) and across country samples (see Supplementary Materials 1). We also refer the reader to Supplementary Materials 3 for more detail about gender reporting.
8. We have ensured that Tables are referred to wherever statements about findings are made in the results.
9. We report the results of power analyses in both the Methods (lines 478-499) and Limitations (lines 1106-1108).
10. We have added the sample size for all models to each table of results.

Responses to Reviewer 1 Comments

- 3. I appreciate the authors’ work in revising the paper. I think it has improved. Yet, I also continue to have some concerns that I believe need to be addressed prior to publication.***

We wish to thank Reviewer 1 for their positive response and thoughtful engagement with the revised manuscript. We greatly appreciate their constructive feedback and the opportunity to further improve our paper.

- 4.***
- a. First, I continue to think the introduction is confusing. The authors start with affective polarization and then discuss perceived polarization. Then they introduce their measure. They never compare any of these measures (also note there is work that looks at perceptions of the other side’s affective polarization – see Lees and Cikara NHB, Less et al. PNAS Nexus). I would strongly encourage the authors to substantially shorten the introduction and more succinctly make the point that they are introducing a distinct measure. It is an interesting measure, and it can stand on its own. Right now, it instead just confuses things.***

We appreciate the reviewer’s helpful comments about the Introduction. In response, we have substantially revised and streamlined this part of the manuscript to clarify the distinct contribution of our approach. Specifically, we now introduce the perceived climate of polarization earlier in the Introduction and provide a more concise explanation of the construct’s focus and advantages. We have also substantially reduced and reorganized our discussion of existing affective polarization research, which may have previously contributed to confusion.

We thank the reviewer for highlighting the relevance of Lees and Cikara's work on meta-perceptions related to affective polarization. We fully acknowledge the conceptual overlap of this work, which also examines perceptions of intergroup political dynamics. However, we elected not to incorporate this literature into the Introduction because it centers on partisan perspectives, whereas our approach focuses on observations of partisan dynamics, regardless of political identification. We believed that including this research could risk conflating these distinct conceptual approaches and blur the focus of our approach. In keeping with the reviewer's recommendation to present the contribution clearly and concisely, we instead focused on highlighting our conceptual approach as directly and coherently as possible.

The revised Introduction now reads (lines 57-319):

“In 2024 and 2025, voters in more than 70 countries around the world participated in national elections that will reshape global politics for years to come. In many of these countries, there are concerns about deepening political divisions as societies seemingly devolve into adversarial political factions (McCoy et al., 2018; McCoy, 2023; Nord et al., 2025). Escalating concerns about political polarization has motivated a growing body of literature investigating its origins and consequences. This research primarily focuses on two forms of polarization: ideological polarization, or the distance between partisans' policy or issue-based positions, and affective polarization, or the deep-seated animosity, distrust, and avoidance between political groups (Lees and Cikara, 2021). Findings underscore the harmful effects of affective polarization, in particular, which may undermine social cohesion, erode trust in democratic institutions, and hinder effective policy development and governance (Iyengar et al., 2012; Iyengar et al., 2019; Torcal & Carty, 2022; Wilson et al., 2020; Hetherington & Rudolph, 2015). Recent work has also identified the specific conditions under which partisan animosity shapes political behaviors (Druckman et al., 2024). Yet relatively less attention has been given to how people perceive antagonistic dynamics between political groups, or the broader conditions that correspond with those perceptions.

The current study examines the perceived climate of polarization, which captures the degree to which people believe major voter groups in their society dislike, distrust, and distance themselves from one another. Rather than assessing personal feelings toward political outgroups, this perspective aims to understand how people perceive the broader relational dynamics between partisan groups in their society. This approach builds on research exploring perceived polarization, which has primarily focused on perceptions of ideological divisions. Past research suggests individuals often overestimate the ideological distance and demographic differences between voter groups in society (Ahler & Sood, 2018; Levendusky & Malhotra, 2016) and these misperceptions can motivate more extreme policy positions (Ahler, 2014) and amplify dislike and distrust for political opponents (Druckman et al., 2022; Enders & Armaly, 2019). These findings suggest that perceived polarization – the psychological experience of polarization – may shape how people think and behave within polarized contexts. By shifting the focus from perceptions of ideological divisions to perceptions of partisan antipathy and avoidance, our approach opens further avenues for understanding how people make sense of political polarization in society.

Addressing perceptions of the societal climate rather than personal animus also contributes to existing literature on affective polarization in two important ways. First,

this approach allows us to explore how affective polarization is experienced by everyone in society, including political independents. Even individuals who do not personally express partisan bias may shape, experience, and be influenced by antagonistic dynamics between voter groups. Yet political independents are often omitted or reassigned a partisan identity in research relying on feeling thermometers that capture negative evaluations of political outgroups (e.g., Kingzette et al., 2021; Druckman & Levendusky, 2019; Reiljan, 2020; Boxell et al., 2020). Second, this approach facilitates more inclusive multinational research by overcoming the challenges associated with aggregating partisan like-dislike scores in multiparty systems, where the number and structure of political groups can vary widely (see Torcal & Comellas, 2025; Wagner, 2021; and Reiljan, 2020 for a more detailed discussion). By accommodating both non-partisan perspectives and diverse political contexts, the perceived climate of polarization offers opportunities to further explore the social dimensions of polarization around the world.

How might broader societal conditions relate to perceptions of a polarized societal climate? One relevant lens comes from research on anomie, which captures the collective perception that society has become disintegrated and dysregulated due to weakened social cohesion and ineffective institutions (Durkheim, 1897; Teymoori et al., 2016, 2017). In a 28-country multinational study, Teymoori and colleagues (2016) found that societal instability was associated with heightened perceptions of breakdown in the social fabric and leadership. We propose that such perceptions can influence how people interpret and organize their social world. When people see society as unstable, they may adopt simpler, more structured interpretations of their social environment to regain a sense of predictability and control (Landau et al., 2015; Kirkland et al., 2024; Teymoori et al., 2017). Establishing clear-cut political categories, in particular, can offer people a straightforward heuristic for anticipating others' values, beliefs, and behaviors. However, this process may also amplify perceived group differences (Tajfel & Turner, 1979; Turner et al., 1987), reinforcing a sense that political camps inhabit separate moral and social worlds. Indeed, prior research shows that perceptions of anomie are associated with lower perceived community cohesion (Teymoori et al., 2017). In this way, perceptions of anomie may serve as a mechanism linking societal instability to heightened perceptions of partisan antipathy and distance. Testing these relationships could contribute a more nuanced understanding of how macro-level features of society relate to the perceived climate of polarization.

The Current Study

Using a multinational dataset spanning 44 countries, the current study aimed to explore the societal conditions associated with a stronger perceived climate of polarization. A secondary goal was to consider whether perceptions of anomie could represent a psychological pathway linking societal conditions to people's impressions of the broader sociopolitical climate. To this end, we conducted exploratory analyses testing the direct relationships between dysfunctional and disruptive country-level conditions and the perceived climate of polarization. We then examined whether the same societal conditions are related to perceptions of anomie – specifically, a perceived breakdown in the leadership and social fabric. These analyses clarify whether the macro-level indicators correspond with subjective impressions of societal instability. We also evaluated whether perceptions of leadership and social fabric breakdown directly relate to the perceived climate of polarization. Finally, we conducted exploratory mediation analyses to assess whether perceived anomie may help explain links between adverse societal conditions

and the perceived climate of polarization. As these analyses were conducted using observational data with unmanipulated mediators, they do not rule out alternative causal pathways and should be interpreted as a preliminary exploration of potential mechanisms rather than evidence of causality (Bullock et al., 2010)."

- b. I also continue to find the statement about political independents misleading – in prior work, it would be only 12% that would not be included but as written it almost suggests 43% would be excluded. This is misleading. There is no need for that. Just make the point that pure independents are typically excluded.***

We thank the reviewer for this feedback. As suggested, we have now removed this framing from the manuscript. Instead, we state (lines 91-95):

"Even individuals who do not personally express partisan bias may shape, experience, and be influenced by antagonistic dynamics between voter groups. Yet political independents are often omitted or reassigned a partisan identity in research relying on feeling thermometers that capture negative evaluations of political outgroups (e.g., Kingzette et al., 2021; Druckman & Levendusky, 2019; Reiljan, 2020; Boxell et al., 2020)."

- c. Finally, if they want to tie affective polarization to functionality, they should see Druckman et al.'s 2024 book.***

We appreciate the reviewer's suggestion to consider Druckman et al. (2024). We now reference this work in the opening paragraph to acknowledge its contribution to understanding the conditions under which partisanship shapes political behavior. Specifically, we state (lines 65-70):

"Findings underscore the harmful effects of affective polarization, in particular, which may undermine social cohesion, erode trust in democratic institutions, and hinder effective policy development and governance (Iyengar et al., 2012; Iyengar et al., 2019; Torcal & Carty, 2022; Wilson et al., 2020; Hetherington & Rudolph, 2015). Recent work has also identified the specific conditions under which partisan animosity shapes political behaviors (Druckman et al., 2024)."

We decided not to elaborate on this literature in greater depth because its focus on partisanship-driven behavioral outcomes falls beyond the scope of our study. Although we touch on the consequences of affective polarization in the opening paragraph to contextualize the broader issue, our primary aim is to examine the societal conditions that correspond with a stronger perceived climate of polarization. To streamline the Introduction, we have tried to limit our discussion to material that more directly clarifies this objective.

- 5. Second, the theoretical discussion that connects anomie to the measures is too abstract. More is needed to introduce the precise measures.***

We thank the reviewer for this helpful suggestion. In response, we have revised our description of the Perceptions of Anomie scale to more clearly articulate why it is an appropriate and informative choice for the present study. In particular, we emphasize how the inclusion of both subscales – Perceived Breakdown of Leadership and Perceived

Breakdown of Social Fabric – enables us to examine whether dysfunctional and disruptive societal conditions relate not only to dissatisfaction with institutions but also perceived erosion in the social foundations that structure everyday interactions. This approach offers a more informative lens for assessing whether and how societal conditions may correspond with a perceived climate of polarization, compared with measures that focus more narrowly on institutional evaluations. The revised text is located on lines 413-426:

“Both subscales were included to capture people’s broader interpretations of societal conditions. Although related measures such as external efficacy and trust in government may be more common in the literature, they emphasize narrower judgments about institutional responsiveness and legitimacy. In contrast, the Perceptions of Anomie scale encompasses evaluations of institutional breakdown alongside perceptions of erosion in the social fabric. The social fabric subscale reflects beliefs that moral standards are weak, that people are uncooperative and untrustworthy, and that interactions are not governed by shared norms. This broader perspective allows us to assess whether dysfunctional and disruptive societal conditions correspond with a diffuse sense of societal breakdown, beyond just discontent toward institutions. The Perceptions of Anomie scale therefore provides a more informative lens for examining whether and how societal conditions may relate to the perceived climate of polarization.”

- 6. Third, the description of the data is incomplete. The authors state that it came from Dynata as a convenience sample and then in the appendix they report the gender distributions. Were quotas used in drawing samples? Were weights applied? If neither quotas nor weights were used I would be uncomfortable with the presentation of the country distributions. Those could easily be inaccurate due to the nature of the samples and readers may be misled in their comparative assessments of countries. I am less worried about the correlational analyses that use such samples but making points about precise estimates if the samples are indeed virtually pure convenience samples seems problematic to me.***

We thank the reviewer for this feedback and acknowledge this oversight. We now provide a fuller overview of the demographic details for the full sample in Table 2 (line 506) and present a breakdown of sample characteristics by country in Supplementary Materials 1. In the Methods, we also clarify the sampling procedure (lines 340-345):

“Dynata recruited participants through pre-established panels in each country, aiming for an equal gender split and a broad age distribution among adults aged 18 and older. However, strict quotas and population weights were not applied. Consequently, the samples should not be considered nationally representative. More information about the samples of each country can be found in Supplementary Materials 1 and 2.”

In line with the reviewer’s concerns, we no longer compare or interpret country-level mean scores in the revised manuscript. We have removed Figure 1 from the Results, which previously presented country-level distributions of Perceived Climate of Polarization scores. Additionally, we have removed all previous comparisons of country-level means from the Results and Discussion. We hope these revisions adequately address the reviewer’s concerns about non-representative sampling.

7. ***Fourth, I do not understand why the authors do not directly add the anomie measures to Table 2 analyses. This seems like a missed opportunity. The current Table 2 is not particularly interesting. It also is confusing to say that coefficients smaller than .15 should be interpreted with caution when most of the coefficients are less than .15. It does not follow from a power analysis to ignore coefficients.***

We thank Reviewer 1 for this helpful suggestion. In response, we have now incorporated the direct associations between each country-level indicator and the perceived anomie subscales into the table (now Table 4, line 645). We agree that this addition provides a more complete picture of how societal conditions relate not only to the perceived climate of polarization, but also to perceptions of breakdown in the leadership and social fabric of society.

We also appreciate the reviewer's concern about the interpretation of small coefficients. In the Results, we now aim to place greater emphasis on the broader pattern of associations across country-level indicators, while still transparently contextualizing the magnitude of individual coefficients. This approach better reflects the exploratory nature of the analyses and their focus on identifying relationships between the perception-based measures and broader dimensions of societal conditions. Specifically, we have revised the relevant section of the Results as follows (lines 585-632):

“As detailed in Table 4, the LMMs featuring country-level indicators found a stronger perceived climate of polarization in countries with less favorable economic conditions (lower GDP PPP per capita; greater income inequality, unemployment, and youth NEET), as well as heightened violence and conflict (greater conflict and instability, political violence, and homicide). Governance-related indicators were the strongest correlates: a one-standard deviation increase in government effectiveness corresponded with a 0.20-point lower score on the perceived climate of polarization, with similar associations observed for corruption control (-0.18) and rule of law (-0.17). For youth NEET and conflict and instability, a one-standard deviation increase was associated with a ~0.15-point higher score on the perceived climate of polarization. Relatively weaker, but statistically significant, associations were also observed for GDP PPP per capita, income inequality, homicide, and political violence.

Indicators of the digital media landscape were also significantly associated with perceived polarization, though effect sizes were smaller in magnitude ($|\beta| < 0.15$). Lower consistency in online media and higher internet and social media use were each associated with a stronger perceived climate of polarization. By contrast, environmental and public health indicators (droughts, floods, and extreme temperatures; pathogen prevalence; food insecurity; childhood mortality; and life expectancy) and democratic strength showed non-significant and near-zero relationships.

As shown in Table 4, many of the country-level indicators associated with the perceived climate of polarization were also significantly related to perceptions of anomie, with more consistent associations observed for the social fabric subscale. Higher income inequality, unemployment, and youth NEET were positively related to both anomie subscales, with unemployment showing the strongest association with perceived breakdown of leadership ($\beta = -0.20, p = .001$) and social fabric ($\beta = 0.17, p < .001$). Higher homicide rates were also associated with both leadership ($\beta = 0.19, p = .003$) and social fabric breakdown (β

= 0.14, $p = .002$), whereas political violence exhibited relatively weaker ($|\beta| < 0.15$), though significant, relationships.

Of the governance indicators, only government effectiveness and corruption control were consistently related to both anomie subscales; weaker political stability and rule of law were only significantly associated with stronger perceptions of social fabric breakdown. Similar results emerged for the digital media landscape: online media consistency was significantly associated with both subscales, whereas internet and social media use showed weaker but significant associations with only social fabric breakdown ($|\beta| < 0.15$). By contrast, environmental and public health indicators showed an inconsistent pattern of associations with perceived leadership and social fabric breakdown, and democratic strength was not significantly related to either anomie subscale.”

For ease of interpretation and in line with feedback from Reviewer 3 (comment 15), we flag coefficients with $|\beta| < 0.15$ in Table 4 using the symbol “^”. We explain in the table notes that these coefficients were estimated with lower precision given the available power. We hope these revisions strike an appropriate balance the concerns expressed by Reviewers 1 and 3.

8. ***I realize the authors want to treat anomie as a mediator but that leads to my fifth point. Adding a reference to mediation problems in the limitation section is insufficient. The mediation analysis does not tell us very much as it could easily be the case that their dependent variable is shaping anomie or some other variable is. Right now, the mediation analysis is almost the centerpiece of the paper and they are not particularly credible. Instead, I would add the anomie measures as correlates in Table 2. I am fine with them including the mediation analysis but it should be stated at the outset that they are at best exploratory.***

We would like to thank the reviewer for their suggestion to reconsider how prominently the mediation analyses feature in the manuscript. In response, we have made several substantial revisions.

First, we now clarify in the Introduction that we explore the role of perceptions of anomie as a secondary research goal. Specifically, we state (lines 243-246):

“Using a multinational dataset spanning 44 countries, the current study aimed to explore the societal conditions associated with a stronger perceived climate of polarization. A secondary goal was to consider whether perceptions of anomie could represent a psychological pathway linking societal conditions to people’s impressions of the broader sociopolitical climate.”

Furthermore, we now include new analyses that examine the direct relationships between (1) country-level conditions and the perceived anomie subscales and (2) the perceived anomie subscales and the perceived climate of polarization. We clarify the steps taken for these analyses in the Method of Analysis section (lines 464-469):

“Next, we explored the relationships between country-level indicators and perceptions of anomie, re-running the previous models twice with (1) perceived breakdown of leadership and (2) perceived breakdown of social fabric as the outcome variables. We then tested

whether perceptions of leadership and social fabric breakdown directly related to the perceived climate of polarization in a separate LMM that controlled for demographics.”

We present the results of these analyses in Table 4 and describe them in lines 603-641 of the Results:

“As shown in Table 4, many of the country-level indicators associated with the perceived climate of polarization were also significantly related to perceptions of anomie, with more consistent associations observed for the social fabric subscale. Higher income inequality, unemployment, and youth NEET were positively related to both anomie subscales, with unemployment showing the strongest association with perceived breakdown of leadership ($\beta = -0.20, p = .001$) and social fabric ($\beta = 0.17, p < .001$). Higher homicide rates were also associated with both leadership ($\beta = 0.19, p = .003$) and social fabric breakdown ($\beta = 0.14, p = .002$), whereas political violence exhibited relatively weaker ($|\beta| < 0.15$), though significant, relationships.

Of the governance indicators, only government effectiveness and corruption control were consistently related to both anomie subscales; weaker political stability and rule of law were only significantly associated with stronger perceptions of social fabric breakdown. Similar results emerged for the digital media landscape: online media consistency was significantly associated with both subscales, whereas internet and social media use showed weaker but significant associations with only social fabric breakdown ($|\beta| < 0.15$). By contrast, environmental and public health indicators showed an inconsistent pattern of associations with perceived leadership and social fabric breakdown, and democratic strength was not significantly related to either anomie subscale.

The broader pattern of results across models suggested that similar macro-level conditions are generally linked to both perceptions of anomie and beliefs about pervasive political divisions. We next examined whether perceptions of anomie were directly associated with the perceived climate of polarization. Results indicated significant positive associations for both anomie subscales when entered simultaneously into the model. Specifically, perceptions of leadership breakdown were positively associated with the perceived climate of polarization ($\beta = 0.07, SE = 0.01, p < .001, 95\% CI [0.05, 0.09]$), while perceptions of social fabric breakdown showed an even stronger positive relationship ($\beta = 0.26, SE = 0.01, p < .001, 95\% CI [0.24, 0.28]$).”

Finally, we have placed less emphasis on the mediation analyses and clarify their exploratory nature throughout the manuscript. We present these analyses as an extension of our exploration of the direct relationships described above. For example:

- *“Finally, we conducted **exploratory** mediation analyses to assess whether perceived anomie may help explain links between adverse societal conditions and the perceived climate of polarization. As these analyses were conducted using observational data with unmanipulated mediators, they do not rule out alternative causal pathways and should be interpreted as a preliminary exploration of potential mechanisms rather than evidence of causality (Bullock et al., 2010).” (lines 253-319)*
- *“Finally, we conducted **exploratory** tests of indirect effects using the lavaan package (Rosseel, 2012) to examine whether the two dimensions of anomie mediated the relationship between each country-level indicator and the perceived climate of polarization.” (lines 470-472)*

- *“Building on these direct associations, we next explored whether perceptions of anomie could help explain the link between country-level conditions and the perceived climate of polarization.”* (lines 657-659)
- *“Taken together, these findings point to perceived anomie as one plausible psychological pathway linking adverse societal conditions to beliefs about partisan antipathy and distance. Results from the exploratory mediation analyses were consistent with this interpretation, though causal direction cannot be established.”* (lines 715-719)
- *“Additionally, our cross-sectional and exploratory design precludes strong conclusions about the directionality of relationships. While our mediation models explore a potential mechanism through which societal dysfunction and disruption may relate to the perceived climate of polarization, they provide no definitive basis for causal inference. As noted by Bullock et al. (2010), mediation models with unmanipulated mediators are vulnerable to biased estimates and cannot account for all possible confounds. The results of our mediation analyses should therefore be interpreted with caution.”* (lines 1055-1063)

9. Sixth, the authors response on WEIRD countries is reasonable but then I would minimize the extent to which that is emphasized – it is odd to emphasize it and then not explore it.

We thank the reviewer for this thoughtful follow-up comment. We have now removed this framing from the manuscript and instead make broader references to the value of a diverse sample. For example:

- *“By accommodating both non-partisan perspectives and diverse political contexts, the perceived climate of polarization offers opportunities to further explore the social dimensions of polarization around the world.”* (lines 219-222)
- *“Moreover, we have presented the first cross-national analysis of how people perceive the climate of partisan animosity in their society, using a large and diverse multinational dataset that includes the perspectives of partisans and non-partisans alike.”* (lines 816-819)

10. In sum, I continue to find the paper interesting and an important contribution. I think addressing my concerns expressed here would not be difficult. I am afraid if the authors do not address them/revise, I cannot support publication as I worry the paper as written could be misleading (e.g., in comparing across countries, the relationship with work on affective polarization, mediation analyses). I would like to see this published and think it would be easy to do. And be a very valuable contribution. The authors may simply disagree with my concerns which is fine but it would leave me not supportive.

We appreciate the reviewer’s continued engagement with our manuscript and believe that their feedback has played an invaluable role in improving this work. We are also grateful for the reviewer’s recognition of the contribution that this paper could make. In response to the reviewer’s concerns, we have made substantial revisions throughout the paper. In particular, we have tried to ensure that the revised manuscript directly responds to the issues raised regarding the relationship with existing literature, cross-country comparisons, and mediation analyses. We hope these changes demonstrate our genuine

effort to constructively engage with the reviewer's feedback and our commitment to presenting the strongest and clearest version of this research.

Responses to Reviewer 2 Comments

- 11. The authors have done a nice job with their revisions. Although I still have some disagreements with respect to interpretation and also think that causal direction is awfully hard to know, I think the study is impressive in the breadth of its coverage and execution and ought to be published.***

We thank Reviewer 2 for their positive and thoughtful assessment of our revisions. We are pleased that the reviewer now finds the manuscript suitable for publication and sincerely appreciate the time and care that they have invested in strengthening this work.

Responses to Reviewer 3 Comments

- 12. I have read the revised version. While I think the authors have addressed many of my concerns. Three issues remain. I leave it up to the editor to decide what to do with them.***

We thank the reviewer for revisiting the revised manuscript and for recognising the progress made in addressing earlier concerns. We appreciate the additional points raised and respond to each below. We are grateful for the reviewer's continued engagement, which has substantially contributed to improving the clarity and rigour of the manuscript.

- 13. To be honest, I do not see convincing evidence for the validity of the used measure in terms of construct validity. Is this indeed a measure of perceived polarization. I leave it up to the editors to decide if this is an issue or not. For me, I would like to see more evidence that the outcome measure we have in this study captures the concept the authors claim to study.***

We appreciate the reviewer's concern about construct validity for the Perceived Climate of Polarization scale. In response, we conducted additional analyses to evaluate how the scale relates to three theoretically relevant constructs: Perceived Moral Polarization and Perceived Moral Overlap (both perception-based scales published by Crimston et al., 2021) and Affective Polarization (measured using feeling thermometers). These variables were included in the broader dataset, allowing us to test convergent and discriminant validity.

Across pooled individual-level, between-country, and within-country analyses, the Perceived Climate of Polarization scale was more strongly related to other perception-based constructs than to affective polarization. This pattern supports convergent validity (alignment with other perception measures) and discriminant validity (weaker relationship with partisan bias). One measure (Perceived Moral Overlap) showed a weaker within-country association than expected. However, this discrepancy was limited to a single variable at one level of analysis. Overall, the broader pattern of results provides support that the Perceived Climate of Polarization scale is more strongly related to *perceptions* of societal polarization than to expressions of partisan bias. We report item

wording for the other scales and detail the findings of these analyses in Supplementary Materials 4.

14. The problem with mediation analyses with your design is that you cannot trust the estimates as they are biased in unknown ways. See the work by Bullock & Green.

We thank Reviewer 3 for their feedback, which echoes concerns raised by Reviewer 1 (Comments 7 and 8). We have now revised the manuscript to place greater emphasis on the direct relationships between the perceived anomie subscales and (1) country-level indicators and (2) the perceived climate of polarization. We also further highlight the exploratory nature of the mediation analyses throughout the manuscript. Please refer to our responses to Comments 7 and 8 for more detail.

15. Thanks for the sensitivity analysis. If $\beta = .15$ and larger are sufficiently powered, then I think you should be careful putting any weight on smaller associations. I think the manuscript and visuals could do a better job helping the reader to draw the correct conclusions.

We wish to thank the reviewer for this feedback and agree that effect sizes and statistical power should inform the weight placed on individual associations. In revising the manuscript, we aimed to highlight the overall pattern of results across models while avoiding overinterpretation of smaller associations that were less reliably detected given the available power. We now place greater emphasis on stronger and more consistent relationships and contextualize weaker associations within the broader pattern of results. In line with Reviewer 3's feedback, we also visually indicate smaller coefficients in Table 4 with the “^” symbol.

We hope these revisions adequately address the concerns expressed by Reviewer 3, while also attending to the feedback provided by Reviewer 1 in Comment 7. Please refer to our response to Comment 7 for more detail about the revisions made.

2 February 2026

Resubmission – Manuscript # COMMSPSYCHOL-25-0433

Country-level instability is related to a stronger perceived climate of polarization across 44 countries

Dear Dr Philipp Schmid and Reviewers,

Thank you for your ongoing engagement with the manuscript. We are delighted to submit a revised and formatted version for your consideration.

Below, we address the reviewers' latest comments and explain the revisions made in response. Please note that all references to line numbers pertain to the manuscript document with tracked changes enabled.

Thank you once again for your thoughtful feedback throughout the review process, which has substantially improved the manuscript.

Responses to Reviewer 1 Comments

- 1. I appreciate the authors careful response to my comments. I support publications. I have a few thoughts, however.***

We wish to thank Reviewer 1 for their positive response and for their helpful suggestions throughout the review process.

- 2. The changes to the introduction are very good. I see why the authors might not want to get too engaged with the work by Lees and Cikara and Lees et al. Yet, not acknowledging them at all may give readers the impression that you are unaware of the work. If footnotes are allowed, I would simply add a footnote at the end of the second paragraph that says there is work that looks at perceived dislike by the other partisan side (Lees and Cikara, Lees et al.) You differ from this work by looking at perceptions of the system more generally.***

We thank Reviewer 1 for this suggestion. In response, we now reference work by Lees and Cikara and clarify how our approach differs in the second paragraph of the Introduction (lines 65-76). As *Communications Psychology* does permit the use of footnotes, we incorporated this text into the main manuscript:

“This approach builds on research exploring perceived polarization, which has primarily focused on perceptions of ideological divisions. Past research suggests individuals often overestimate the ideological distance and demographic differences between voter groups in society^{11,12} and these misperceptions can motivate more extreme policy positions¹³ and amplify dislike and distrust for political opponents^{14,15}. Some research has also examined beliefs about political outgroup animosity and distrust, capturing perceptions of affective divisions from a partisan perspective^{16,4}. Together, this literature suggests that perceived polarization – the psychological experience of polarization – may shape how people think

and behave within polarized contexts. By shifting the focus from partisan perspectives to broader perceptions of antipathy and avoidance between political groups, our approach opens further avenues for understanding how people make sense of political polarization in society.”

- 3. *The text suggests as line 345 that direct effects of anomie are entered into the model but they do not appear in Table 4 -- so I am confused. The response memo is not helpful b/c the line numbering is distinct from what I see (it likely changed when uploaded). I think somewhere there should be a multi-variate model that readers can see with those variables entered.***

We wish to thank Reviewer 1 for highlighting this lack of clarity. Table 4 presents the results of the linear mixed models examining associations between the country-level indicators and the three perception-based variables: (1) perceived climate of polarization, (2) perceived breakdown of leadership, and (3) perceived breakdown of social fabric. The latter two variables correspond with the two dimensions of the Perceptions of Anomie scale.

We estimated an additional model examining the direct relationship between the two anomie subscales and the perceived climate of polarization. However, we did not include this model in Table 4 because the table is already large and focused on the associations between country-level indicators and the perception-based variables. Instead, we reported the results of the model examining perceived anomie and perceived polarization in the main text (lines 346-352):

“We next examined whether perceptions of anomie were directly associated with the perceived climate of polarization in a separate model. Results indicated significant positive associations for both anomie subscales when entered simultaneously into the model. Specifically, perceptions of leadership breakdown were positively associated with the perceived climate of polarization ($\beta = 0.07$, $SE = 0.01$, $p < .001$, 95% CI [0.05, 0.09]), while perceptions of social fabric breakdown showed an even stronger positive relationship ($\beta = 0.26$, $SE = 0.01$, $p < .001$, 95% CI [0.24, 0.28]).”

To improve clarity, we have restructured the Results section. We now first report all models examining relationships between the country-level indicators and perception-based variables (lines 307-342), followed by Table 4, which presents the full results of those models. We then report the direct relationships between the two anomie subscales and the perceived climate of polarization, clarifying that these were examined in a separate model (lines 344-352). We hope these revisions reduce ambiguity about the distinction between these sets of models.

Responses to Reviewer 3 Comment

- 4. *Thank you, I think you have addressed my comments.***

We thank Reviewer 3 for their positive response and thoughtful engagement with the manuscript throughout the review process.